# Thermal-comfort evaluation of and plan for public space of Maling Village, Henan, China

**Qindong Fan[1], Fengtian Du[2]\*, Hu Li[1], Chenming Zhang[1]**

**1** School of Architecture, North China University of Water Resources and Electric Power, Zhengzhou, China,
**2** School of Art and Design, North China University of Water Resources and Electric Power, Zhengzhou, China

\* dufengtian@126.com

**Data Availability Statement:** All relevant data are within the manuscript and its Supporting Information files.

**Funding:** This study was supported by 2019 Key R&D and Promotion Projects in Henan Province (No. 192102310004), Henan Ecological Civilization

## Abstract

The thermal environment of village public space affects the comfort of people ' s outdoor activities, and then affects the willingness of residents to outdoor activities, which has an important impact on the villagers' quality of life. Previously published studies of thermal comfort mostly focused on the evaluation of thermal comfort index, few studies on the application of thermal comfort planning. The study was carried out in Maling Village, Changdai Town, Mengjin County, Luoyang City, Henan Province, China. Square, street, green space were chosen as three typical public spaces where thermal comfort indexes were measured by questionnaire survey and field measurement during summer. Subsequently, the village's microclimate environment was simulated with ArcGIS 10.6 and ENVI-met. The results indicate that during the summer, the influences of temperature, solar radiation, wind speed, and relative humidity on the subjective comfort conditions of the outdoor environment gradually decreased. The spatial form of village has an important influence on thermal comfort. Finally, based on the results, this study put forward the thermal comfort process and planning scheme of the village outdoor space.

## 1. Introduction

Approximately half the world's population lives in rural areas [1], the outdoor spaces of which have great functional importance in the residents' lives. Rapid urbanisation has disturbed the rural outdoor environment [2]. In the past, rural planning has primarily considered spatial layout, culture, ecology, and other village functions [3, 4]. As such, the relationship between a village's spatial form and microclimate has most neglected. The state of thermal comfort in the village outdoor space is bad, which has lowered villagers' willingness of outdoor activities [5–7].

Thermal comfort, as the most direct reflection of the human body to environment, has quickly become a hot spot in the study of microclimate outdoor [8]. Studies of thermal comfort conditions can be traced back to the 1930s [9]. Currently, thermal comfort is defined as the 'condition of mind that expresses satisfaction with the thermal environment and can be assessed by subjective evaluation' [10]. There are three aspects of factors that affect outdoor

City Theory and Application Innovative Scientific and Technological Team, 2020 Henan Science and Technology Think Tank Research Project (HNKJZK-2020-02C and HNKJZK-2020-25C).

**Competing interests:** The authors declare no conflict of interest.

thermal comfort: the physical, personal, and social. Physical factors include air temperature, relative humidity, solar radiation, and wind speed [11]. Personal and social factors include a resident's gender, age, metabolism, residence time, emotional status, cultural background, environmental attitude and cloth conditions [12–14].

Traditional outdoor thermal comfort research is mainly based on field measurement [15–21], including outdoor air temperature, relative humidity, black ball temperature and wind speed and other environmental parameters, as well as the subjective questionnaire survey in the study area [22, 23]. The purpose of field measurements was to determine the thermal conditions that are comfortable or acceptable for local residents, understand the residents' perceptions of thermal comfort [24, 25]. The case studies focus on the livability evaluation of outdoor thermal environment, and the influence of urban form, ventilation and heat island to outdoor thermal comfort [26, 27]. Recently, with the development of science and technology, numerical simulation has become an important means of outdoor thermal environment analysis, and the ENVI-met software has been one of the most widely used platforms [28–30]. ENVI-met is suitable for small- and medium-scale microclimate simulations; it has been used in relevant case studies of hot-summer-cold-winter regions [31], hot-summer-warm-winter regions [29, 30], temperate regions [32, 33], cold regions [34], hot and arid regions [35, 36], and tropical regions [37, 38]. Furthermore, it has been applied to many different outdoor microclimate spaces, such as streets [39], campus areas [40], parks [41], residential communities [42] and green-roofs [43].

The most common thermal-comfort evaluation indexes are the standard effective temperature (SET), predicted mean vote (PMV), PET (physiological equivalent temperature), universal thermal climate index (UTCI), mean radiant temperature ($T_{mrt}$), wet bulb globe temperature (WBGT), and discomfort index (DI) [44, 45]. The PMV is determined by the stable indoor environment [46], and a seven-point scale is often used to predict the average thermal response of many people in indoor thermal-comfort evaluations [47]. The UTCI is an easy-to-determine body heat index, which can be applied to different seasons, climates, and dimensions [48, 49]. Furthermore, the WBGT, SET, $T_{mrt}$, and DI have been widely used to assess the outdoor thermal comfort characteristics of different climate zones [50–52]. The PET, which is more sensitive than UTCI to changes in human thermal response, is a widely studied parameter in thermal-comfort research on weather forecasting, urban planning and design [53–55], and the outdoor evaluation of complex environments in different climate zones [56, 57].

At present, outdoor thermal comfort research is mostly concentrated in urban public space. A large number of scholars have carried out specific research on different public spaces such as squares [58–60], parks [41, 50, 61] and streets [62–64]. The effects of solar radiation, ventilation and human activities on temperature in different public spaces are the main research directions [65–67]. Other scholars studied microclimate based on urban local climate zone [68–74]. Nevertheless, the research of village thermal environment is based on architecture and climatology, and its influence on village thermal environment is mainly analyzed from the perspective of village architectural form [75–78]. A large number of scholars analyze the relationship between buildings and rural climate adaptability from the aspects of building layout [79, 80], form [81], and materials [82]. In general, the research on village thermal comfort mainly focuses on micro-climate environment [83], thermal comfort [84], climate adaptability characteristics [85], energy consumption [86], and the influence of architectural form on thermal comfort [78], etc.

In summary, the research methods and theoretical system of public thermal comfort has been relatively mature. However, the research on thermal comfort of rural public space is still not in-depth, especially the planning research on thermal environment. In this study, the ArcGIS 10.6 platform, ENVI-met 4.4.5 software, and PET index were used to evaluate the thermal

comfort characteristics of outdoor public spaces (i.e., streets, squares, and green space) in Maling Village, Henan, China. Subsequently, the thermal comfort simulation was carried out, and a specific village thermal-comfort planning scheme and process are proposed.

## 2. Materials and methods

### 2.1. Ethics statement

The field studies were conducted in Maling village, Changdai Town, Mengjin County, Luoyang City, Henan Province, China (Fig 1A). The public open space of the village was taken as the study area. This study was supported by the local government and the masses. The questionnaire survey and field measurement did not involve endangered or protected species. For a detailed explanation, please see the S1 File.

### 2.2. Study area

The study area is in a typical hilly village in the west of Changdai Town, Mengjin County, Luoyang City, Henan Province, China (Fig 1A). It belongs to the warm-temperate (Dwa) [87] climate zone, which is hot and rainy during the summer and cold and dry during the winter. The meteorological statistics from 2010 to 2019 in Fig 1B presents the meteorological data from the China Meteorological Information Centre [88]. The land use data of Maling Village originated from China's third land survey, which was conducted in 2018 (Fig 1C).

Maling Village has a total area of 51 ha and a population of 1419. Agriculture is a crucial sector of Maling Village. Furthermore, the village has well-equipped public service facilities, and residents can perform a variety of outdoor activities in street spaces, squares and many other green spaces. The average building height is relatively low (one to three storeys high).

### 2.3. Selection of measurement points

Three typical public spaces (squares, streets, and green space) in Maling Village were chosen as study areas. A total of 15 measurement points were selected. Seven measurement points were on roads (the measurement points were located on the central axes of roads with buildings on both sides), including four east–west streets and three north–south streets. In addition, four measurement points were in the squares and green spaces of the village, respectively. The specific positions of the measurement points are shown in Fig 2. Moreover, Fig 3 lists the current spatial measurement points.

### 2.4. Methods

The specific research process is as follows:

The first step is the assessment of thermal sensation and thermal comfort and the exploration the effect of temporal distribution regularities and microclimate parameters on residents' outdoor activities.

The second step is the measurement of microclimate in the selected public space.

The third step is the evaluation of outdoor thermal comfort based on PET index.

The fourth step is the establishment of the relationship between PET and thermal sensation.

The fifth step is numerical simulation and thermal comfort planning.

The research framework of this paper is shown in Fig 4.

#### 2.4.1. Evaluation of overall thermal sensation and thermal comfort characteristics.

The study was performed during the summer (from June 12th to 17th, 2020), which are all sunny days, the village's roads, squares, and green spaces were visited, and residents doing

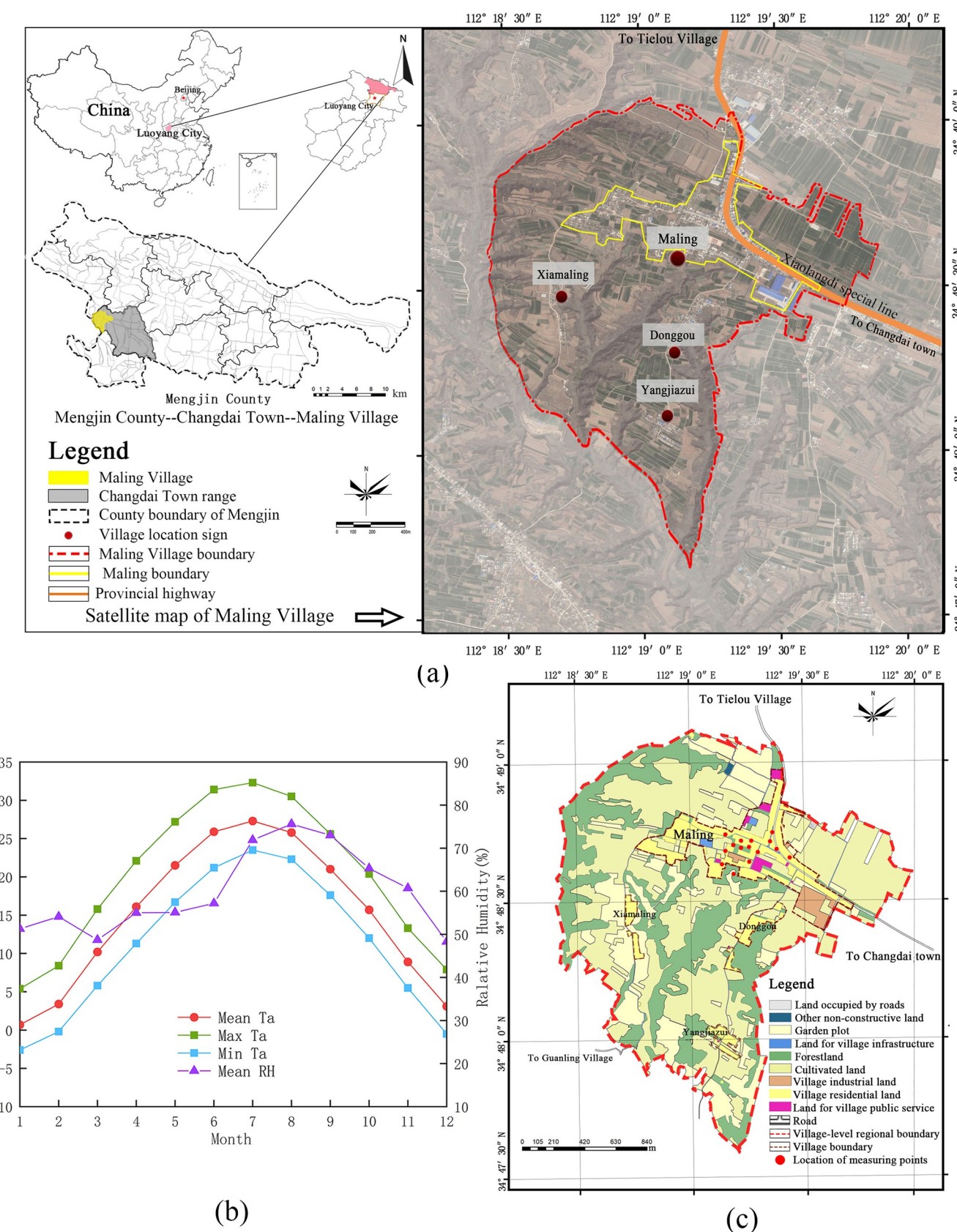

(a)

(b)

(c)

**Fig 1.** (a) Geographic location of the study area. (b) Monthly mean/maximal/minimal air temperature and mean relative humidity in Mengjin County from 2010 to 2019. (c) Land use map. (The satellite images in Fig 1 (a) with permission from Beijing Qianfan Shijing Co., Ltd, original copyright [2020] [96]).

outdoor activities were randomly selected for the survey. 450 local villagers (with ages ranging from 18 to 70; with a 1:1 ratio of men to women) were invited to participate in the survey (Before the interviewees answered the questionnaire, they were asked to sit quietly for 20 min in a shaded area within a 1 min walk from the test site; the purpose of this was to achieve a uniform metabolic rate). Finally, 412 valid questionnaires were obtained. All data were collected through a questionnaire (S1 File), and the subjective data are provided in S2 File.

The questionnaire consist of two parts: basic information about the interviewees and thermal-sensation votes (TSV). Basic information contains gender, age, weight, clothing, types of outdoor activities, and the time they spent outdoors. The TSV contains thermal-sensation and thermal-comfort. Thermal-sensation is presented with the ASHRAE-13 seven-point scale [89], thermal-comfort is presented with five-point scale [61] (Figs 5 and 6). In the questionnaire, the public space were classified into eight categories:

A: space with fitness facilities;

B: unsheltered square;

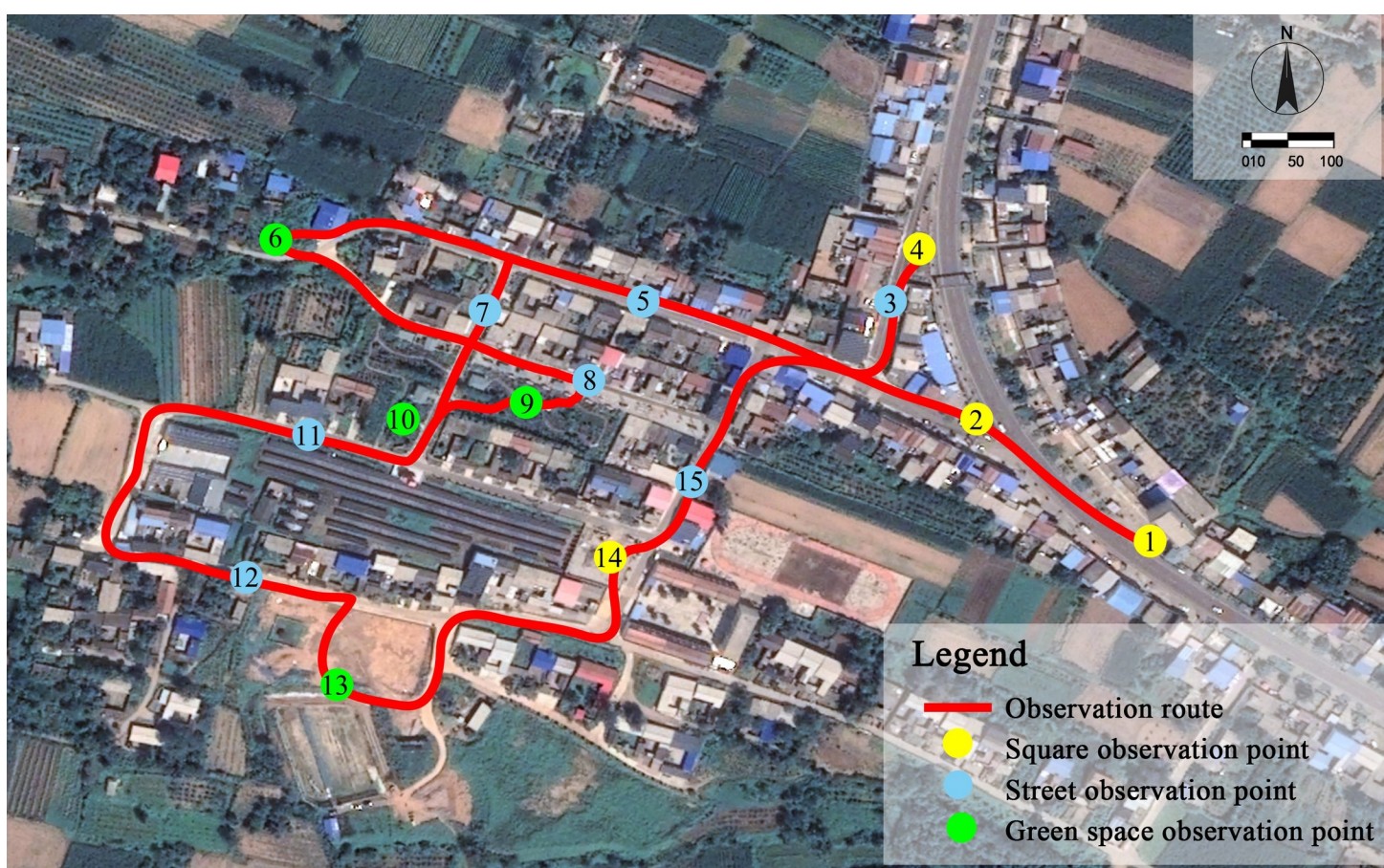

**Fig 2. Distribution of measurement points.** (The satellite images in Fig 2 with permission from Beijing Qianfan Shijing Co., Ltd, original copyright [2020] [96]).

| Square | | | |
|---|---|---|---|
| 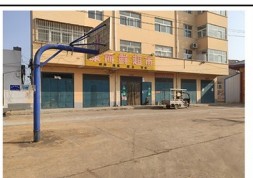 | 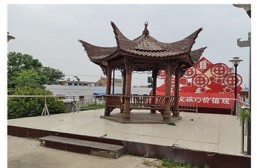 | 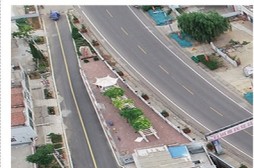 | 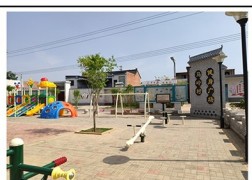 |
| **Description** Point 1, Basketball square, located across the road from the village committee (without shade of plants and buildings). | Point 2, Pavilion square, located on the north side of the East–West Street of Market Road (white hard pavement, containing a pavilion without plants). | Point 4, Triangle square , located on the east side of the North–South Street of Market Road (red hard pavement ), pavilions and corridors, a small amount of shrubs and vines) | Point 14, Fitness square , located on the south side of Photovoltaic Road (mainly red hard pavement including few small trees). |

| East–West Street | | | |
|---|---|---|---|
| 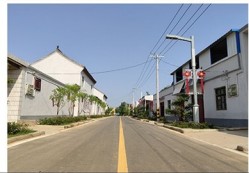 | 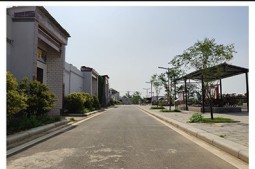 | 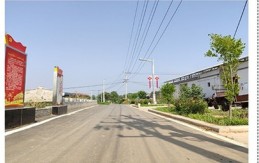 | 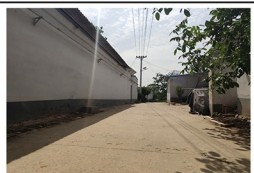 |
| **Description** Point 5, East–West Street of Market Road (asphalt pavement, with street trees on both sides , without building shade ). | Point 8, Maling 5th Group Road (asphalt pavement, with flower beds shrubs and low street trees on both sides, without building shades). | Point 11, Photovoltaic Road (asphalt pavement, with street trees on one side, without building shades). | Point 12, West Street of primary school (concrete pavement, with plants on one side, with building shades). |

| North–South Street | | | |
|---|---|---|---|
| 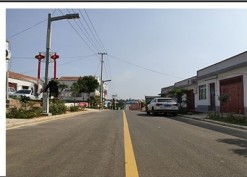 | 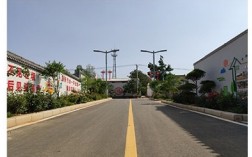 | 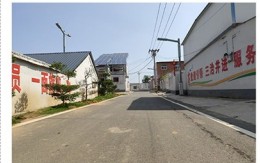 | |
| **Description** Point 3, North–South Street of Market Road (asphalt pavement, some area with no greening and building shades). | Point 7, Park Road (asphalt pavement, with more shrubs on both sides and some building shades). | Point 15, North–South Street of School Road (asphalt pavement, with street trees on one side and some building shades). | |

| Greenland | | | |
|---|---|---|---|
| 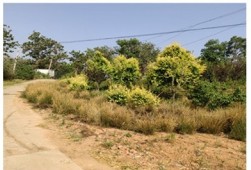 | 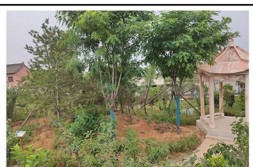 | 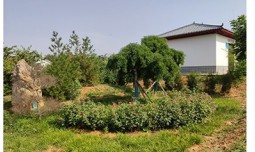 | 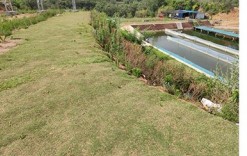 |
| **Description** Point 6, Green space on west side of the village (dominated by shrubs with few grasses). | Point 9, Green space on east side of Park Road (dominated by trees). | Point 10, Green space on west side of Park Road (mixture of shrubs and grass). | Point 13, Green space on south side of the village . (dominated by grass). |

**Fig 3. Different types of spatial measurement points.**

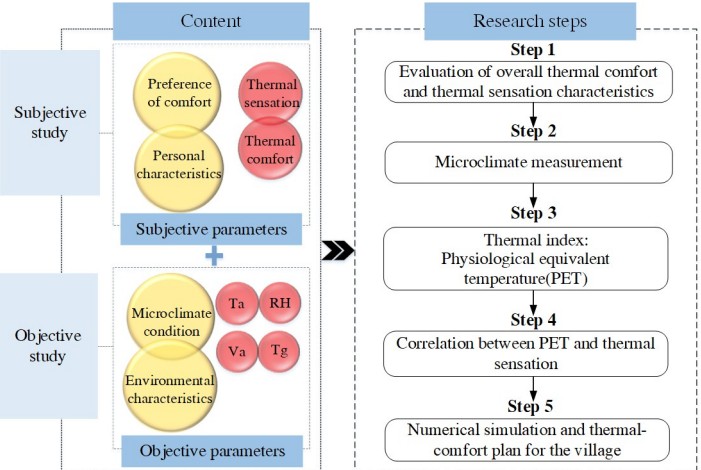

**Fig 4. Research framework for thermal comfort of village public space.**

C: spaces with shading from trees;

D: unshaded lawns;

E: green park spaces;

F: pavilions;

G: areas with shading from buildings;

H: roads

**2.4.2 Microclimate measurement.** The questionnaire study and field measurements were conducted simultaneously; The study was performed during the summer (from June 12th to 17th, 2020); the period covered weekends and working days; data from days with rainly weather condition were excluded. Measurements were made every two hours from 6:00 until 20:00. The measurement time at each sampling point was 5 min.

The selected microclimate indicators, such as the thermal environment parameters of different squares, the varying orientations of streets, and the locations of green spaces, were measured during the selected time period. Based on the measurement and analysis results of three different measurement points at a height of 1.5 m, the data/information of black globe temperature, air temperature, relative humidity, and wind speed at each measurement point were

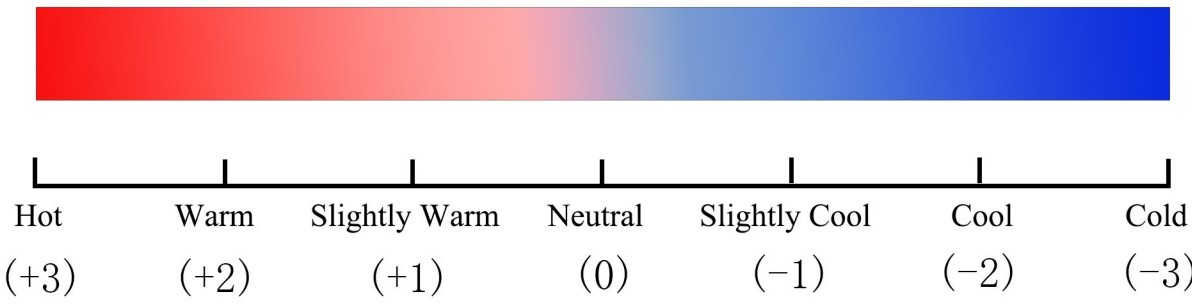

**Fig 5. Thermal-sensation evaluation scales.**

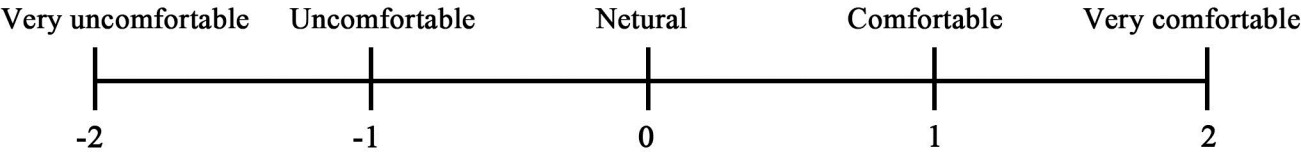

**Fig 6. Thermal-comfort evaluation scales.**

obtained. All data collected are presented in S3 File. The specific measurement instruments are listed in Table 1.

**2.4.3 Thermal comfort indices.** The PET index was chosen based on the subjective data and field measurements to evaluate the outdoor thermal comfort characteristics of the village. The index comprehensively analyses the effects of all physiologically related weather indexes (the air temperature, relative humidity, wind speed, and mean radiation temperature) and personal parameters (the metabolic rate and clothing insulation level). The corresponding thermal sensation and thermal stress (physiological stress) are shown in Table 2 [90].

The RayMan software was applied to calculate the PET [91]. It was assumed that the thermal resistance of clothing (0.9 clo) and active metabolic rate (80 W) remain constant during the calculation, and the physical parameters of the thermal environment were the input parameters (which included the temperature, humidity, wind speed, and mean radiation temperature ($T_{mrt}$)).

For outdoor measurements, the mean radiant temperature can be obtained by approximation from the black globe temperature, air temperature, and wind speed [92]:

$$T_{mrt} = [(Tg + 273)^4 + \frac{1.10 * 10^8 Va^{0.6}}{\varepsilon D^{0.4}}(Tg - Ta)]^{\frac{1}{4}} - 273 \tag{1}$$

where $T_{mrt}$ is the mean radiant temperature; $T_g$ is the black globe temperature (˚C); $T_a$ is the air temperature (˚C); $V_a$ is the wind speed (m/s); D is the globe diameter (0.05 m in this study), and $\varepsilon$ is the emissivity (0.95 for a black globe).

**2.4.4 Correlation between PET and MTSV.** The relationship between the PET and mean TSV of the village's thermal environment during the summer was studied, and the village's neutral temperature range during the summer was determined.

$$MTSV = \frac{\sum_n^j TSV}{n} \tag{2}$$

where n is the number of TSVs of each thermal index, and MTSV represents the mean TSV.

**2.4.5 Numerical simulation and thermal-comfort plan for the village.** Based on field measurement, the ENVI-met model and measurement results were used to simulate the village's microclimate. the ArcGIS 10.6 platform was applied for the analysis, and a thermal-comfort plan that is specific to the village was established. The model area of Maling Village is approximately 30 hm². The sunny June 14th, 2020 was chosen for the simulation. The simulation took

**Table 1. Experimental equipment used for the measurement of micrometeorological parameters.**

| Measured Parameter | Instrument | Range | Accuracy |
|---|---|---|---|
| Air temperature | DT-83 | -20.0~60.0˚C | ±1˚C (0˚C~40˚C) ±2˚C (-20˚C~0˚C&~40˚C~60˚C) |
| Relative humidity | DT-83 | 0.0~100.0% | ±3.5%RH (20%~80%) ±5%RH (0~20%&80~100%) |
| Wind speed | GM8902 | 0~45m/s | ±3%±0.1 |
| Globe temperature | AZ8758 | 0~50˚C | ±3˚C (15~40˚C) |

**Table 2. Classification of thermal sensation/stress on the PET scale.**

| PET (˚C) | >41 | [35, 41) | [29, 35) | [23, 29) | [18, 23) | [13, 18) | [8, 13) | [4, 8) | <4 |
|---|---|---|---|---|---|---|---|---|---|
| **Thermal Sensation** | Very hot | hot | Warm | Slightly warm | Neutral (Comfortable) | Slightly cool | Cool | Cold | Very cold |
| **Thermal Stress** | Extreme heat stress | Strong heat stress | Moderate heat stress | Slight heat stress | No thermal stress | Slight cold stress | Moderate cold stress | Strong cold stress | Extreme cold stress |

24 h (from 06: 00 (GMT + 8), and the simulation intervals were 60 min. Based on the field investigation results, 91 satellite images, and ArcGIS 10.6, the study area was transformed into a base map with BMP format, which can be recognised by ENVI-Met 4.4.5. The map was imported into the sub-module 'SPACE' to establish the model. The maximal building height was set to 17.5 m, and the grid dimensions were defined at $100 \times 60 \times 30$ cells. According to the dimensions and building height of the study area, the grid resolution was set to dx = dy = dz = 2 m, with four nested grids in the peripheral area. The initial microclimate data were the weather data of the macro climate environment on June 14[th], 2020. The dominant wind direction was 170˚; the model roughness length was 0.1, and the underlying surfaces of the model area were an asphalt road and a grey concrete road. The early simulation was based on the actual conditions of the measurement sites, and the later simulation was based on the village's thermal-comfort plan. In addition, the pre-set scheme was used to simulate the microclimate, and the PET indexes were chosen to comprehensively evaluate the village's thermal environment.

## 3. Results and analysis

### 3.1. Subjective evaluation results

**3.1.1. Temporal distribution of outdoor activities and preferences regarding thermal environment parameters.** On working days during the summer, villagers prefer to perform outdoor activities in the morning and evening (mainly between 6:00 and 8:00 and after 16:00) because solar radiation is weaker during those times. The most active time period on weekends is between 18:00 and 20:00 (Table 3) because young people have more time for outdoor activities during evening hours. Therefore, the period during which residents go to public spaces and perform outdoor activities is advanced.

According to the villagers' preferences regarding the thermal environment parameters, the four microclimate parameters' degrees of influence in relation to thermal comfort are as follows: air temperature > solar radiation > wind speed > and relative humidity (Fig 7). 49% of the interviewees stated that temperature had the greatest impact on the their outdoor activities in summer; 24% of the respondents believed that long-term exposure to the summer sun will cause discomfort. In addition, a few of the respondents reported that the summer wind cools

**Table 3. Main time periods for outdoor activities during the summer.**

| Hour | Number of Workdays | Weekend Attendance |
|---|---|---|
| 6:00–8:00 | 65 | 28 |
| 8:00–10:00 | 45 | 57 |
| 10:00–12:00 | 30 | 38 |
| 12:00–14:00 | 10 | 17 |
| 14:00–16:00 | 33 | 30 |
| 16:00–18:00 | 75 | 87 |
| 18:00–20:00 | 88 | 95 |
| 20:00–21:00 | 66 | 60 |

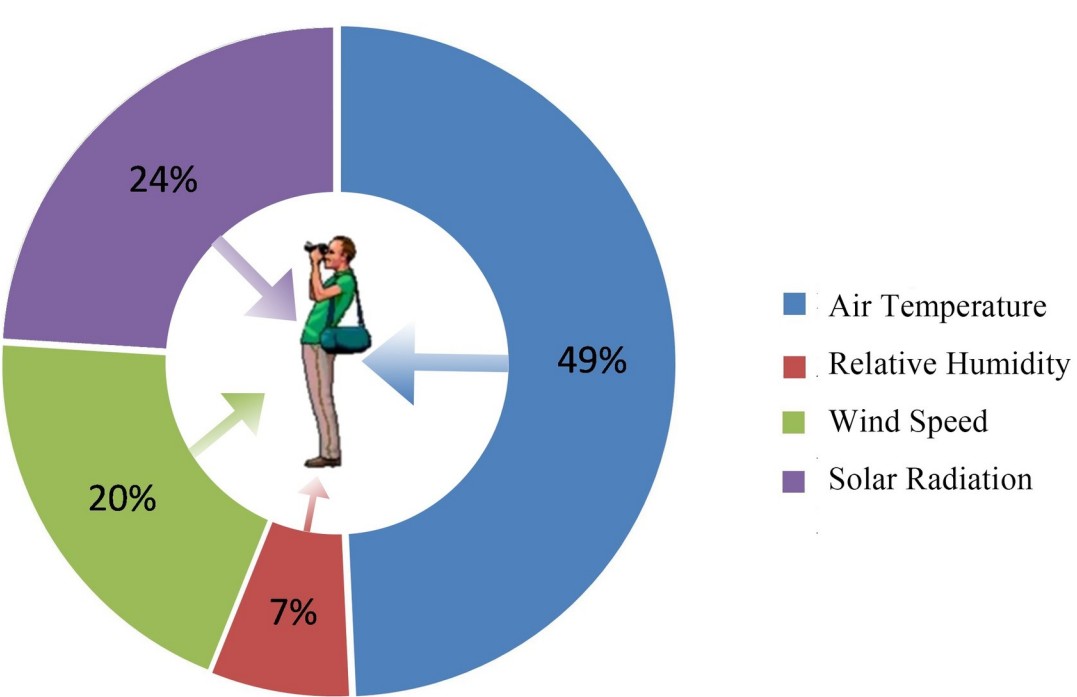

**Fig 7. Evaluation of the influence of certain characteristics on subjective comfort.**

the public spaces, improving thermal comfort characteristics. The majority of respondents reported that they do not think relative humidity significantly affects the thermal comfort level.

**3.1.2. Characteristics of thermal comfort and thermal sensation.** *(1) Evaluation results of overall thermal comfort*. The summer survey results are shown in Fig 8. 67% of the respondents stated that the area shaded by trees has the highest comfort level, followed by the green spaces of parks (51%); 49% of the respondents stated that the thermal environment of unshaded lawns is 'neutral'. In addition, 13% of the respondents stated that the road environment is the lowest in terms of comfort. The village's public spaces in the order of high-to-low thermal-comfort are as follows: green spaces in parks > areas with shading from buildings > artificial structures > unsheltered squares > roads. (A: space with fitness facilities; B: unsheltered square; C: shading from trees; D: unshaded lawns; E: green spaces in parks; F: pavilions; G: areas with shading from buildings; H: roads).

*(2) Overall evaluation results of TSV*. The TSV distribution is shown in Fig 9; the values are mainly distributed in the moderate and over-hot ranges (TSV $\geq$ 0). According to the results, during the test period, the outdoor thermal environment of Maling Village is hot during the summer. The proportion of respondents who felt 'TSV = 0' was the highest; thus, most villagers have adapted well to the thermal environment. Most people reported feeling 'hot' and 'very hot' on the road and 'hot' in the unsheltered square. In addition, the respondents reported different thermal sensations for different types of spaces. This indicates that there are significant differences between different types of spaces in terms of in simultaneous thermal sensations. (A: space with fitness facilities; B: unsheltered square; C: shading from trees; D: unshaded lawns; E: green spaces in parks; F: pavilions; G: areas with shading from buildings; H: roads).

*(3) Mean TSV and PET*. The weighted mean values of the PET and TSV were calculated and fitted with a linear model. As shown in Fig 10, the overall scatter distribution is close to the fitting curve $R^2$ = 0.8105. For a MTSV of 0–0.5, the neutral temperature of the human

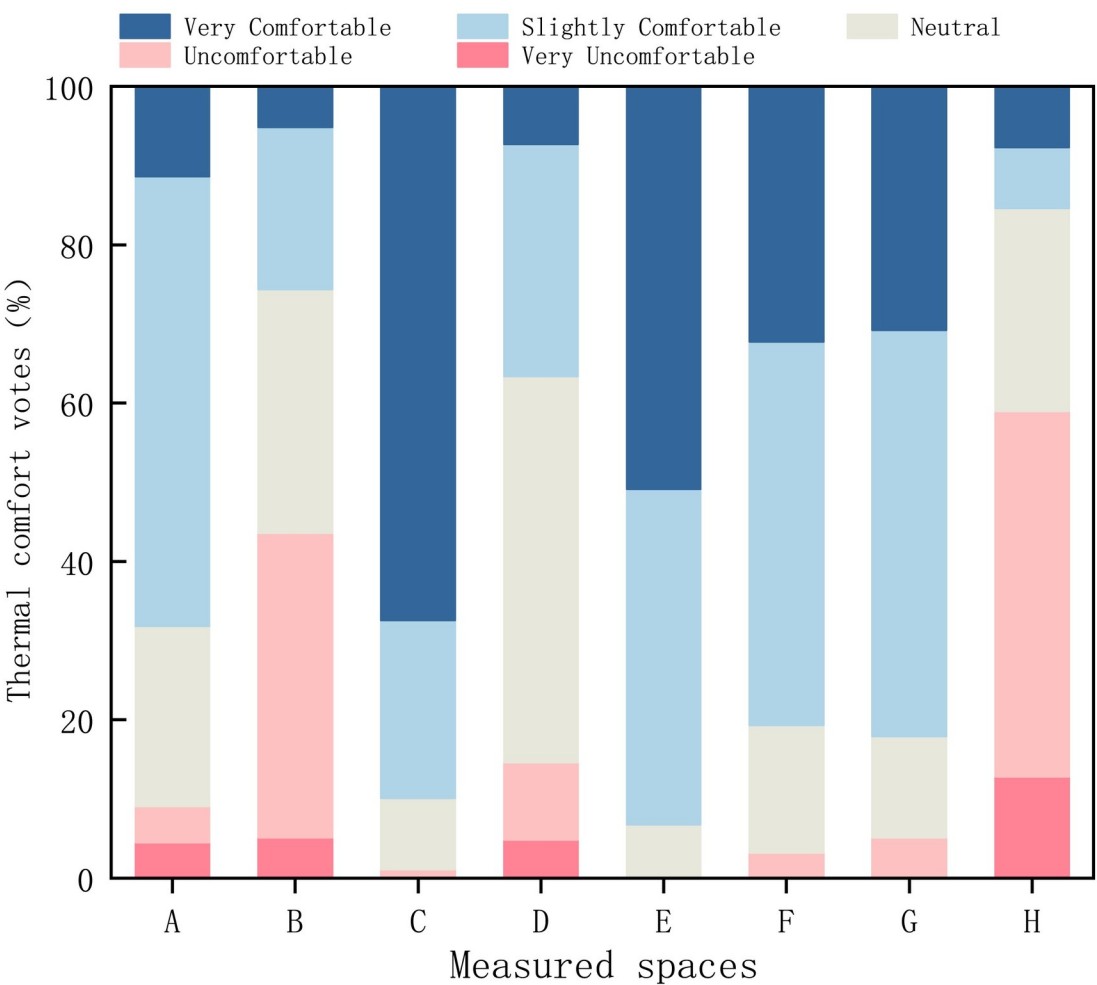

**Fig 8. Overall comfort votes for different spaces.**

body corresponding to the PET is 23.77–27.23°C; the range of the neutral-temperature PET includes 'no heat stress' and 'slight heat stress'. The results reveal a strong correlation between the PET and MTSV, and the degree of fit is good.

## 3.2 Objective evaluation results

**3.2.1. Analysis of various thermal environments.** (1) Measurement and analysis of the village square's thermal environment.

As shown in Figs 11 and 12:

1. Mean air temperature: basketball square (Point 1) > fitness square (Point 14) > triangle square (Point 4) > pavilion square (Point 2).

2. In general, the relative humidity values of the pavilion square (Point 2) and triangle square (Point 4) are higher than those of Points 1 and 14; in addition, the relative humidity and air temperature are negatively correlated.

3. Mean wind speed: triangle square (Point 4) > pavilion square (Point 2) > basketball square (Point 1) > fitness square (Point 14). The wind speed of Point 4 is higher because Point 4

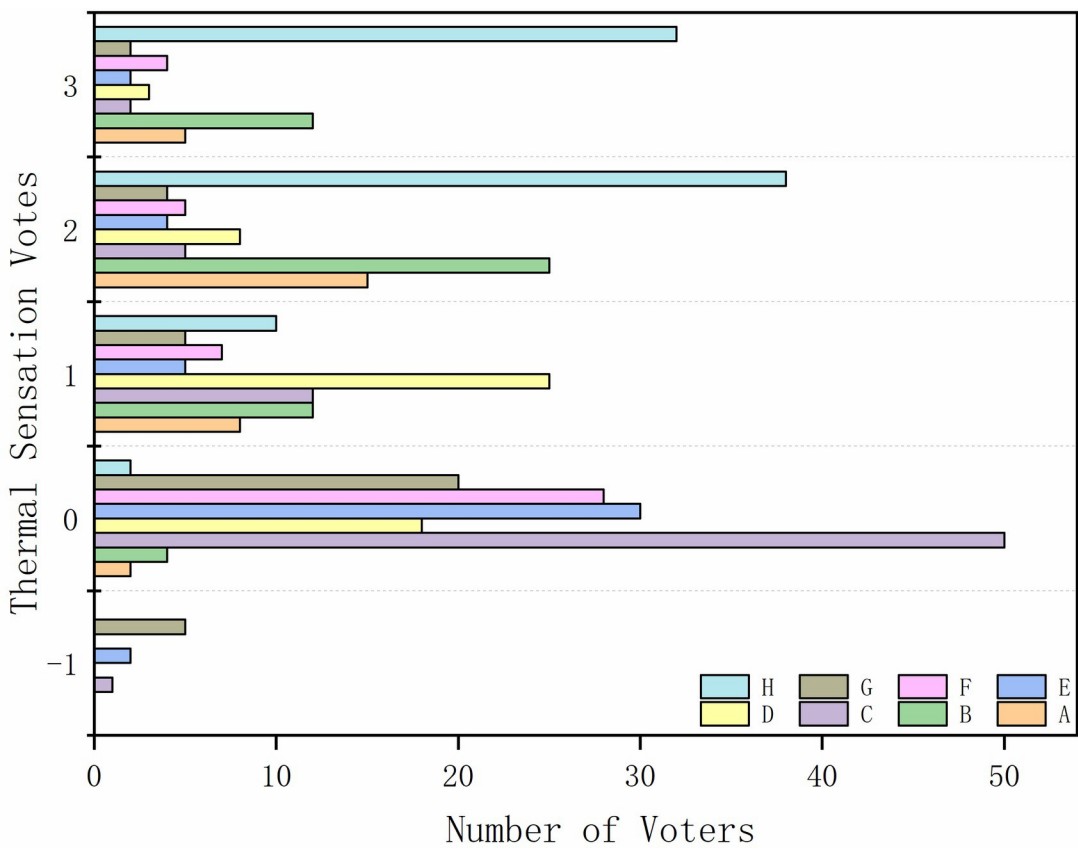

**Fig 9. TSVs for different spaces.**

lies in a sunken square. The east side, which is near the road, is greatly affected by the natural wind direction; by contrast, the wind speed of Point 14 is the lowest because the wind is largely blocked by the building.

4. The black globe temperature at the four measurement points in the square begins to increase after 8:00. Therefore, its trend is consistent with the increase in solar radiation; the change range at Point 1 is the highest, indicating that it is most significantly affected by solar radiation. The black globe temperatures at Points 1 and 14 are high because the points are in an unsheltered square. Measurement Point 14 is only blocked by a few buildings, and Point 1 is open. Furthermore, the square is exposed to solar radiation for an extended period of time during the day. Therefore, the black globe temperatures at these two measurement points are high. The black globe temperature at Point 2 is the lowest and changes slightly; thus, the high buildings, which block the wind and provide shading, significantly influence the black globe temperature.

(2) Measurement and analysis of the village streets' thermal environment.
As can be seen from Figs 13 and 14:

1. Mean air temperature: East–West Street of Market Road (Point 5) > Photovoltaic Road (Point 11) > North–South Street of Market Road (Point 3) > Park Road (Point 7) > Maling 5<sup>th</sup> Group Road (Point 8) > West Street of primary school (Point 12) > North–South Street of School Road (Point 15). The maximal air temperature was measured at Point 5 at

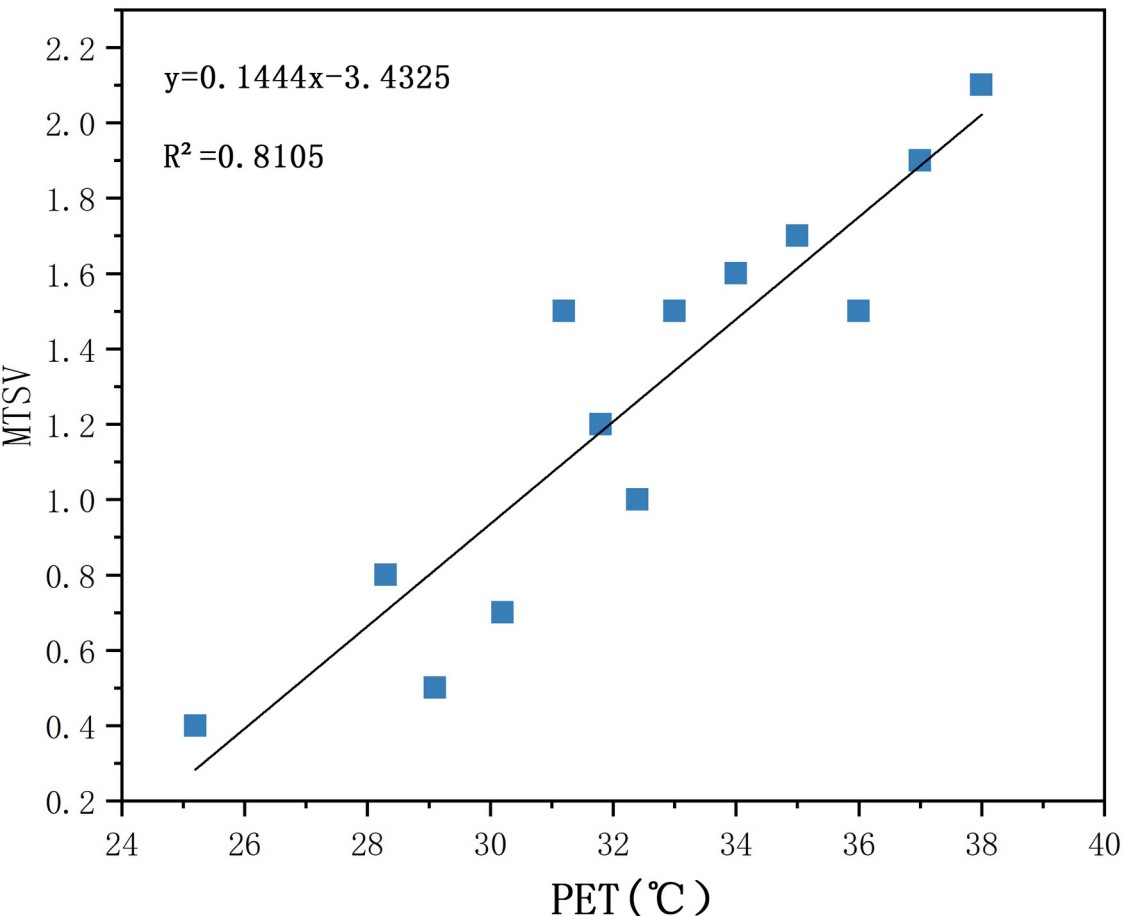

**Fig 10. Correlation between PET and MTSV.**

approximately 14:00 (38.7˚C); during this time, the air temperature at Point 15 was the lowest (34.9˚C; the difference between the maximum and minimum was 3.8˚C). This was due to the building on the east side of Point 15 being higher than that on the west side and providing shading. According to the analysis results of the mean temperature at each measurement point on the East–West street, the mean air temperature at Point 12 is 32.6˚C, which is lower than the temperatures at Points 5, 8, and 11. This difference in temperature is mainly due to the shading from the tall trees lining both sides of the road at Point 12; in contrast, Points 5, 8, and 11 have fewer trees in their vicinity.

2. The maximal black globe temperature of Point 5 was measured at approximately 14:00. (48.1˚C). At this time, the black globe temperature was the lowest (40.9˚C) at Point 15. Hence, there is a difference of 7.2˚C between the maximal and minimal temperatures. In addition, the black globe temperature curve of Points 12 and 15 is relatively stable, and the mean black globe temperature is lower than at other points. The results show that lining a given street with trees and buildings can effectively reduce the impact of solar radiation

3. The hourly variation curve of the wind speed showed no evident trend during the entire measurement process due to the significant, random variations in wind direction. However, when the building layouts are relatively consistent, the fluctuation in wind speed is relatively low for a street lined with trees. Therefore, plants above a certain height significantly reduce the wind speed and improve thermal comfort for a village street.

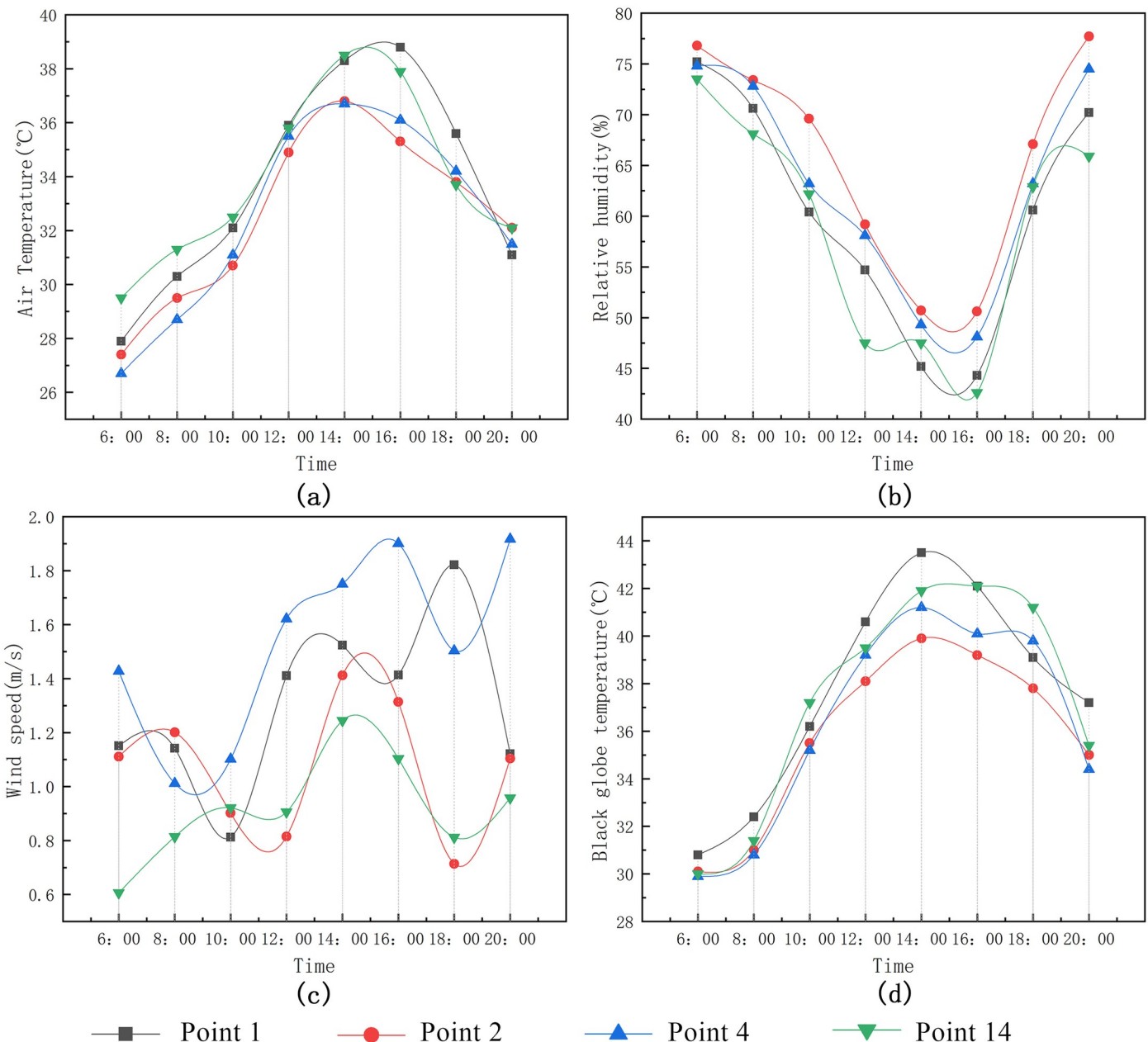

**Fig 11. Variation trends of microclimate parameters at each measurement point in the square.**

4. Overall, the mean relative humidity is highest at Point 15, followed by Point 12. Since there are buildings and trees that provide shading for Points 15 and 12, respectively, the mean relative humidity is relatively high. By contrast, the unsheltered streets are greatly affected by solar radiation during the day; this leads to a low mean relative humidity of the surrounding air.

(3) Measurement and analysis of the village green spaces' thermal environments.
As shown in Figs 15 and 16:

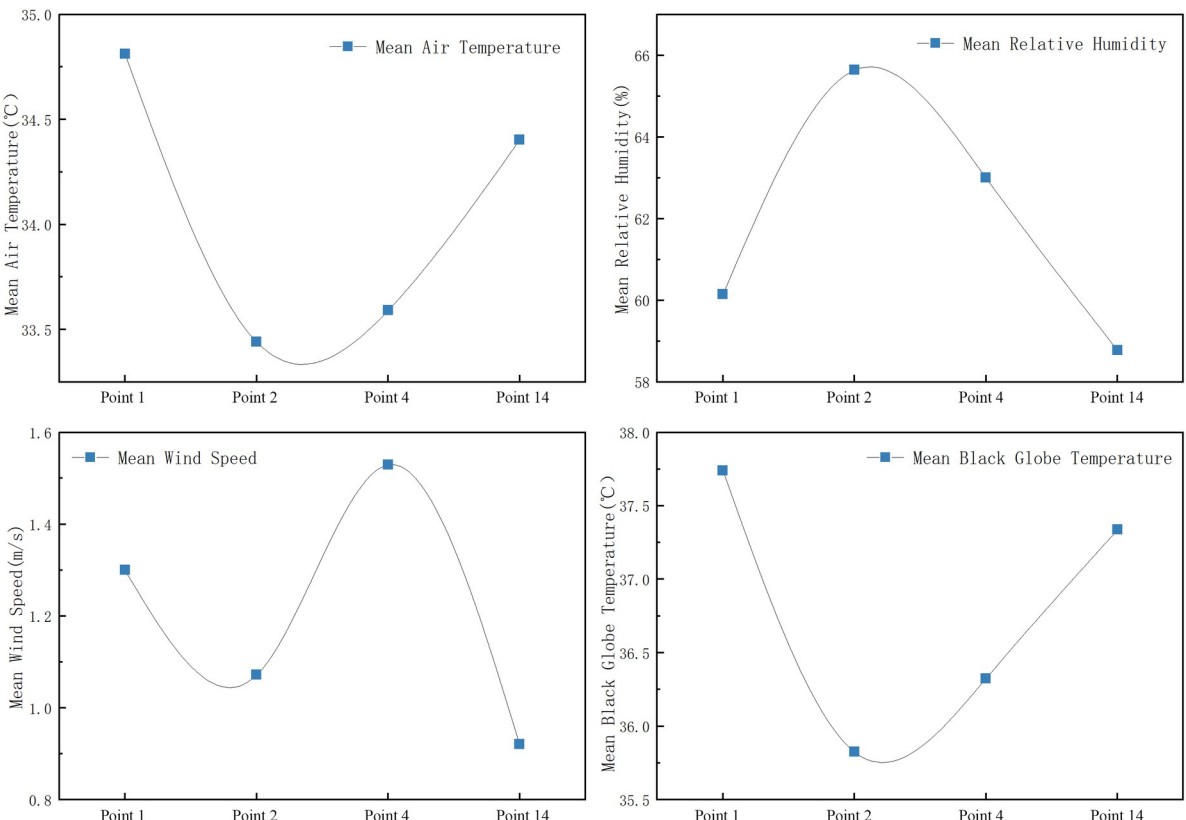

**Fig 12. Mean variation of meteorological parameters at each measurement point in the square.**

1. The order of the mean air temperature is as follows: green space on the south side of the village (Point 13) > green space on the west side of the village (Point 6) > green space on the west side of Park Road (Point 10) > green space on the east side of Park Road (Point 9). The air temperature increases continuously after 8:00 and reaches its maximum at 14:00. The air temperature at Point 13 reaches 36.8˚C, and the air temperature at Point 9 is the lowest (31.9˚C, which corresponds to a difference of 4.9˚C).

2. The air temperature values at Points 9 and 10 gradually increased from 10:00 to 14:00. The change curve is relatively stable because the village employed special personnel responsible for watering and maintaining the plants in the park during this period. The ejected water lowers the air temperature. This effect is similar to direct evaporative cooling. Therefore, during this period, the air temperature of the park was lower than those of the other two measurement points; the maximal difference is approximately 2.7˚C. Hence, human intervention has a short-term and rapid effect on humidity. The change trend of the relative humidity of the green space in the village is inversely proportional to the air temperature: the higher the air temperature, the lower the relative humidity. For example, there are many tree species at Points 9 and 10; their transpiration can increase the relative humidity of the surrounding air. In addition, the air humidity at each green-space measurement point were higher in the mornings and evenings.

3. The highest black globe temperature is measured at Point 13 at 14:00 (46.1˚C). The corresponding black globe temperatures of the two green spaces in the park are the lowest (below

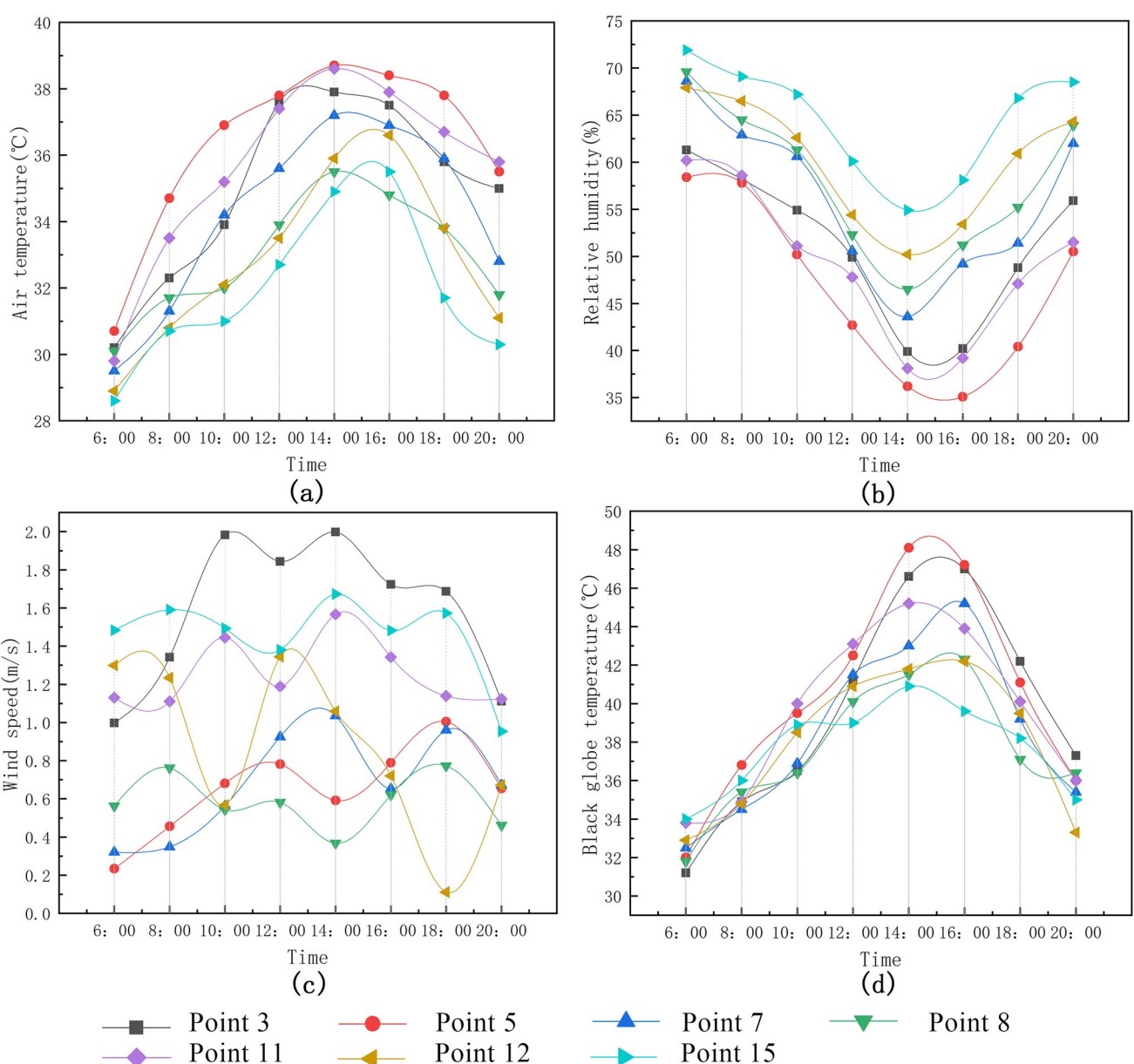

**Fig 13. Variation trend of microclimate parameters at each measurement point in the street.**

40˚C). This is mainly due to the fact that the plants at Point 13 are wild and of the low ground-cover type; the area is primarily covered by unsheltered grassland without planted trees.

4. Points 9 and 10 are surrounded by trees and buildings. Therefore, the wind speeds are lower and more stable at these two points than at the other two measurements points.

**3.2.2 Comparison of thermal environments.**    The mean values of three different space climate impact parameters were studied to compare the corresponding thermal environments; Table 4 presents the results.

The following conclusions can be drawn:

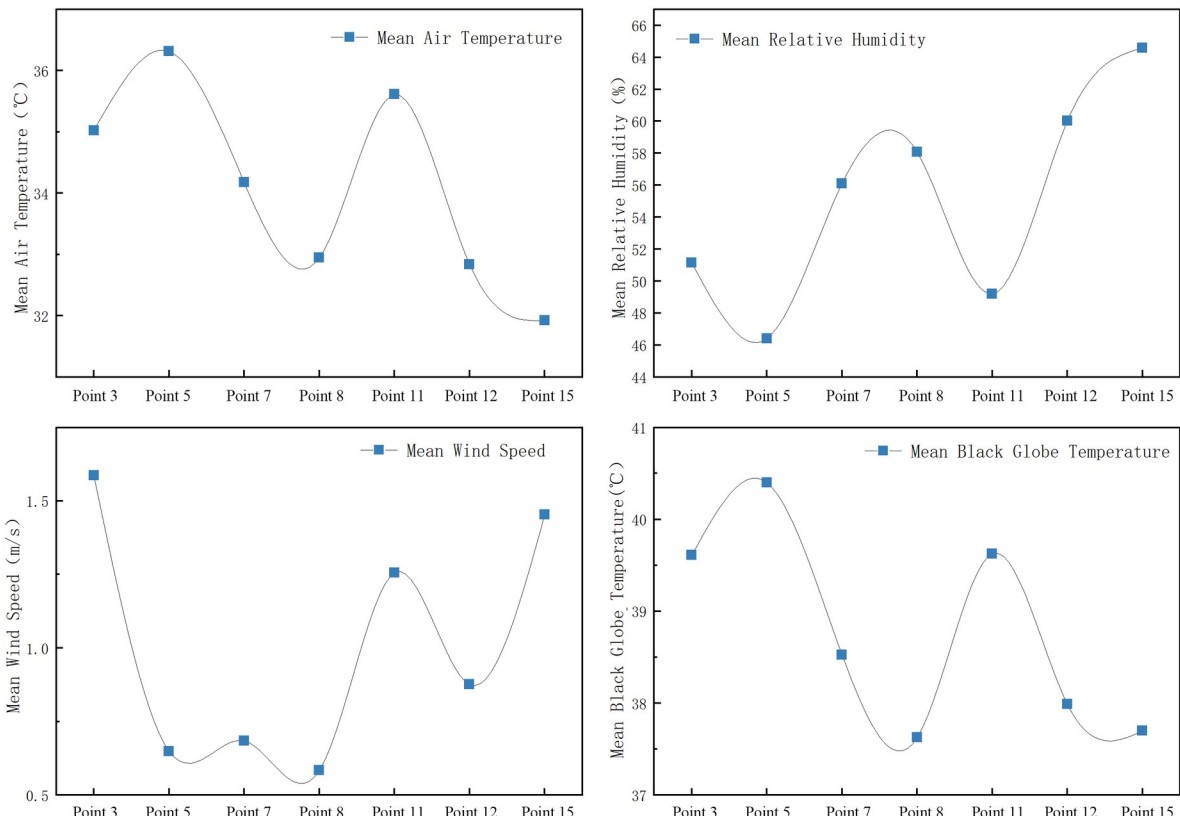

**Fig 14. Mean variation of meteorological parameters at each measurement point in the street.**

(1) The temperature of the green space of the village is lower than the temperatures of the roads and squares; the highest difference is 3.55˚C. This difference is due to the shade, which can effectively reduce the influence of solar radiation, and the transpiration of plants, which absorb the surrounding heat. The watering of plants also helps to cool the area.

(2) The mean relative humidity values of the squares and roads are lower than those of the green spaces; more specifically, the relative humidity of the road is the lowest (approximately 17.85% lower than that of the green space of the village). The village road and square have hard underlying surfaces; most of the roads are made of asphalt pavement, whereas some have been constructed with cement pavement; the square is mainly constructed of white ceramic tile and red square bricks, while the underlying surface of the green space is primarily grassland. The ground of the road is exposed to sunlight for an extended period of time and consists of asphalt and cement pavement; this causes the road to heat up quickly, which reduces the relative humidity in the air. Nevertheless, the evaporation of water and the transpiration of plants increase the air humidity in the village park, significantly increasing the relative humidity.

(3) According to Table 4, the mean black globe temperature of the village road is (approximately 4.58˚C) higher than that of the green space. With the exception of temperature values at Points 12 and 15 on the village roads, the black globe temperature values of the other roads are high due to the lack of street trees and shading from buildings on both sides. In addition, the village has a large green area with dense trees, which effectively

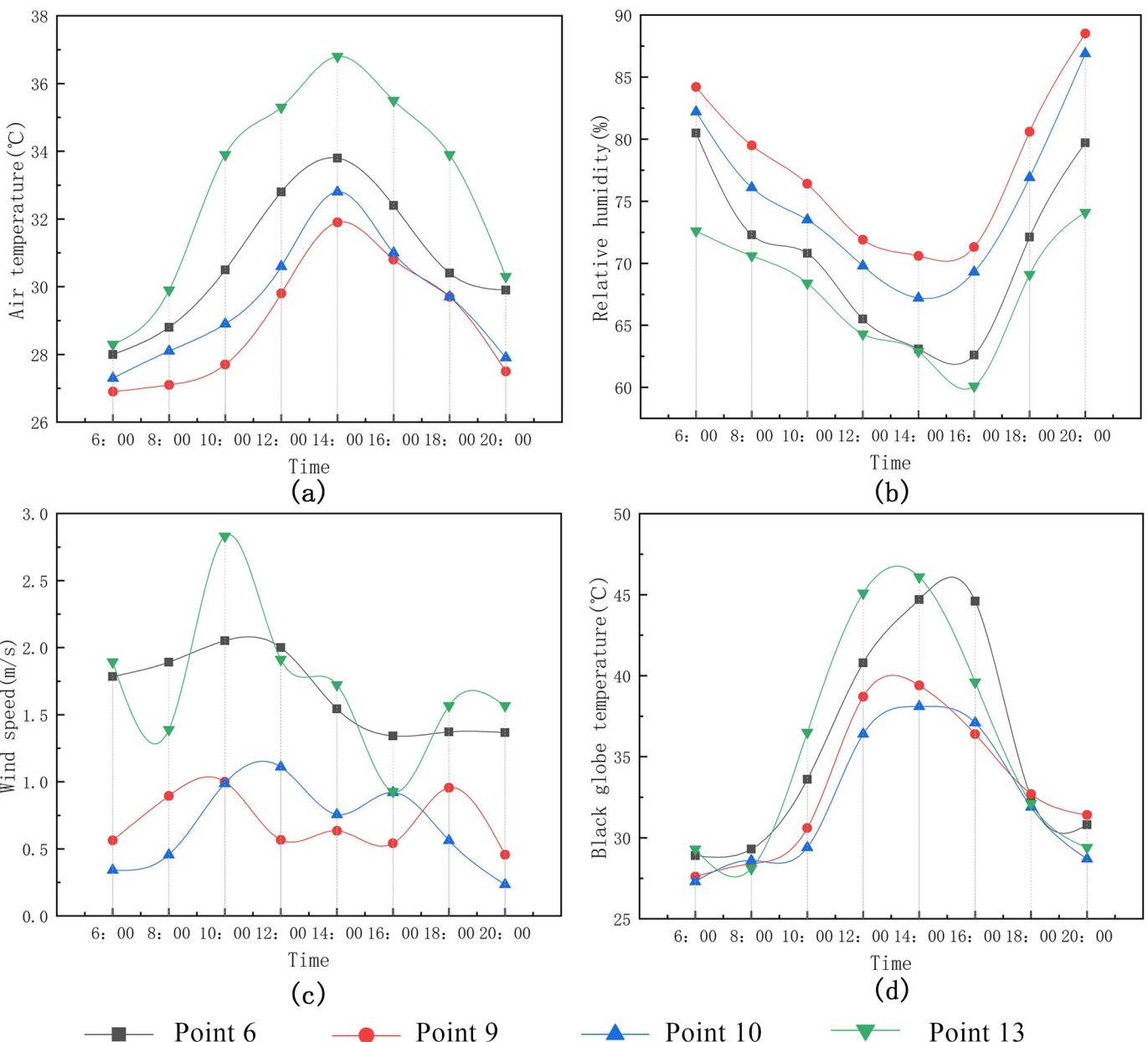

**Fig 15. Variation trends of microclimate parameters at each measurement point in green space.**

block the sunlight and reduce the mean black globe temperature. Consequently, the mean black globe temperature of the village's green space is much lower than that of the road.

(4) The mean wind speeds of each space type exhibited little deviations without evident regularity. In addition, the wind speed at the square exceeded the mean wind speed. As most measurement points are located at an intersection or on the roadside, the wind speed cannot be significantly reduced by trees or buildings.

**3.2.3 Simulation of factors that affect thermal comfort.** The dominant wind direction of the village was south–east on June 14[th], 2020. The wind speed distribution within the study

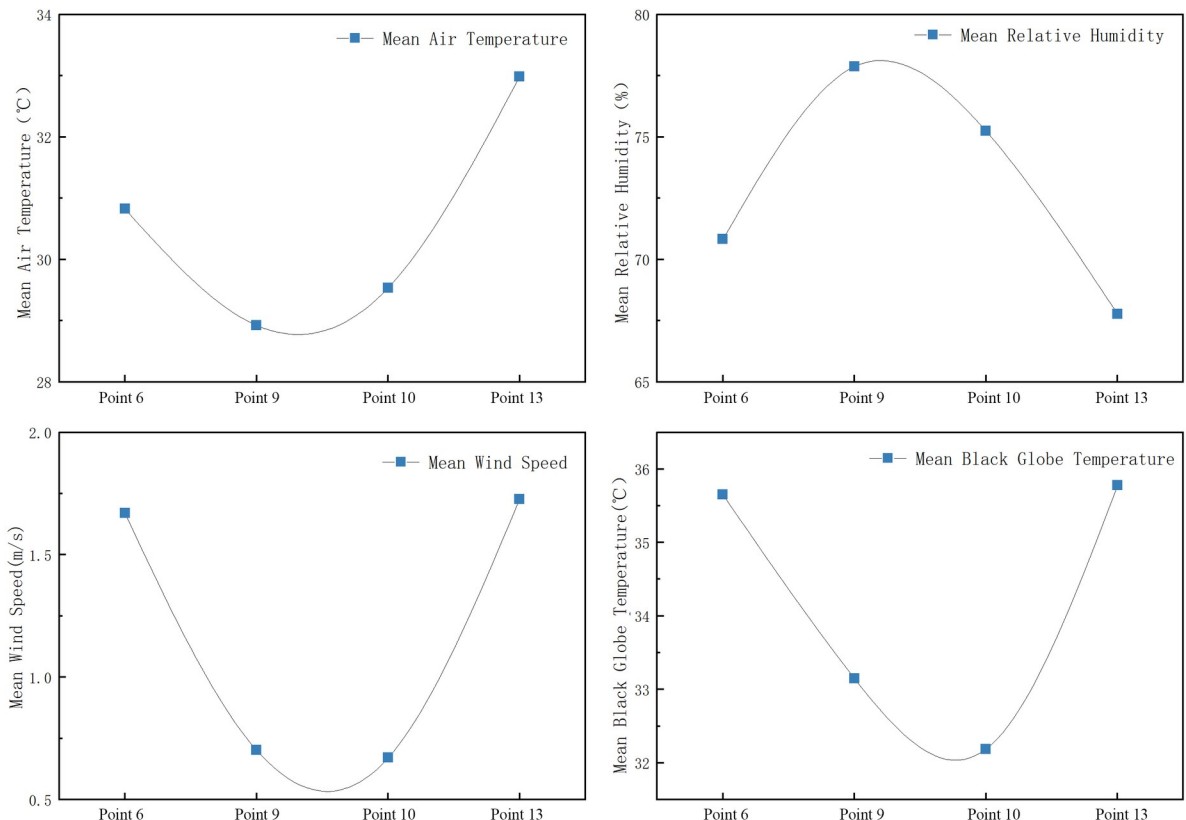

**Fig 16. Mean variation of meteorological parameters at each measurement point of green space.**

area is shown in Fig 17A; the wind speed was generally 0.00–2.45 m/s. For some open public spaces in the middle of the study area, the wind speed reached 0.54–1.36 m/s. Due to canyon effect, the highest wind speed reach between 1.36 and 2.45 m/s at the east side of the village, where buildings are tall. Moreover, the wind speeds at the south and north of the village are greater (1.36–1.63 m/s); this is primarily because the areas are not blocked by buildings. The terrain on the south side is high; the site is spacious; and the overall space is well ventilated. In addition, the solar radiation intensity around the village buildings remains in a comfortable range (indicated by the blue colour). However, owing to the village's low buildings and the relatively sparse plants, the solar radiation intensities of the village road and square are high at 14:00.

The simulation map of the air temperature and relative humidity in the village is shown in Fig 17C and 17D. The temperature of the study area increases gradually from 30.2 to 38.2°C (from dark blue to red). Owing to the tall buildings in the east of the village and the relatively open layout, the temperature is the lowest in this area. This value is followed by that of the green area in the middle of the village (green and light-yellow zones in the map). This area

**Table 4. Mean values of the microclimate parameters of different types of spaces.**

| Space Type | Mean Air Temperature (°C) | Mean Relative Humidity (%) | Mean Black Globe Temperature (°C) | Mean Wind Speed(m/s) |
|---|---|---|---|---|
| Village square | 34.06 | 61.89 | 36.81 | 1.21 |
| Village street | 34.12 | 55.08 | 38.78 | 1.01 |
| Village green space | 30.57 | 72.93 | 34.20 | 1.19 |

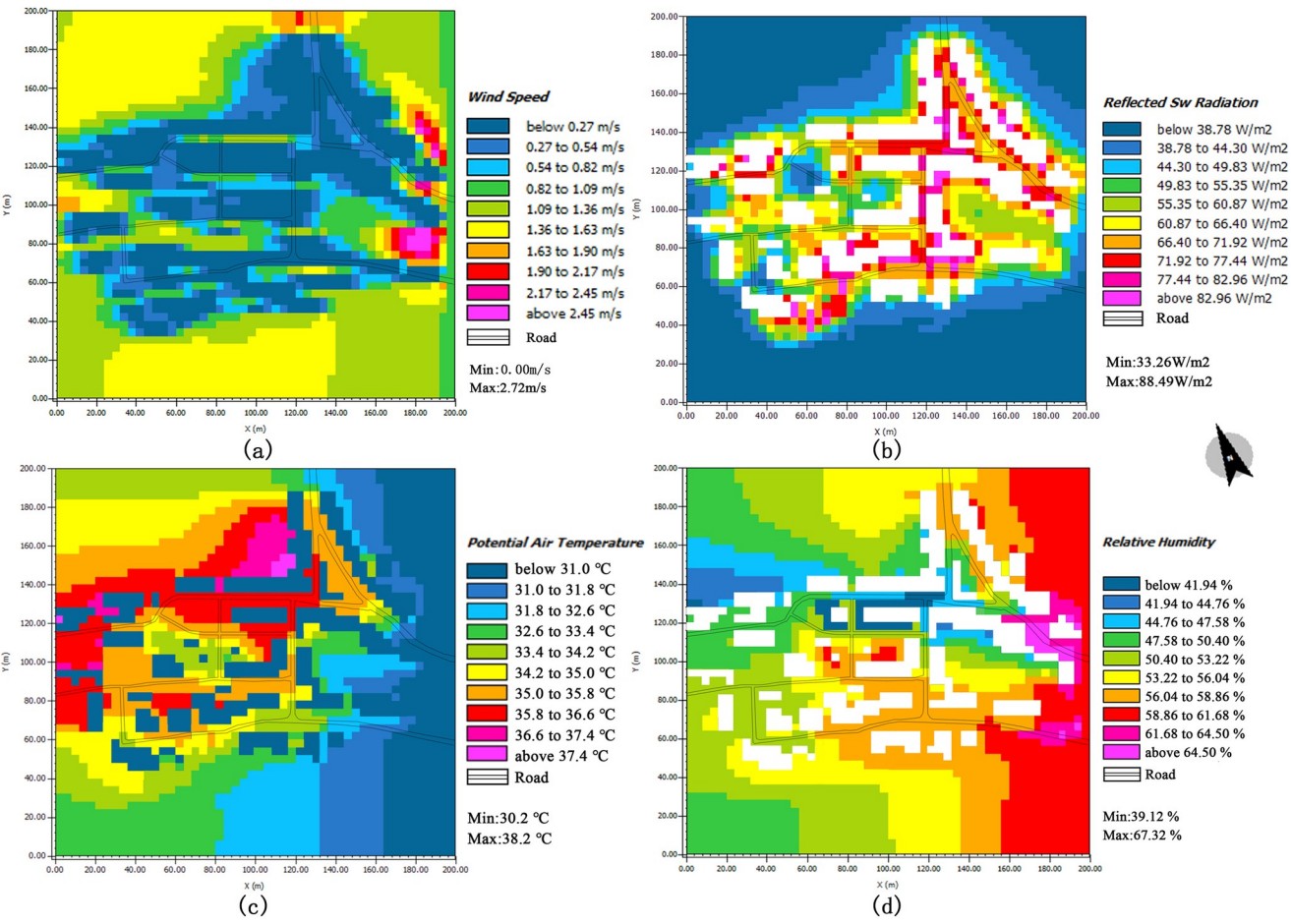

**Fig 17. Simulation results of the thermal environment.**

includes a lot of green space, which has an evident cooling effect at 14:00. In addition, the temperature values of the north and west sides of the village are the highest, whereas the temperature values in the east and south of the denser areas are lower. Therefore, shading provided by buildings and trees can reduce the ambient temperature. The overall humidity of the village was found to range from 35.52% to 80.32%. According to the current conditions, the east side of the village has much open space, rich vegetation, and high air mobility, all of which lead to a high relative humidity. In contrast, due to the high building density on the streets of Market Road, the hard pavement of the public space, and poor vegetation, the relative humidity is the lowest.

**3.2.4 Thermal-comfort plan for the village.** From the site measurements, microclimate simulations, and current land use map of the village (Fig 1C), it can be concluded that the factors affecting thermal comfort are the geographical location, nature of the underlying surface, plant configuration, types of outdoor activities, and building layout, among others. The aim of a thermal-comfort plan is to mitigate the deficiencies of public spaces. The thermal comfort level and planning approaches for the outdoor space of the village are listed in Table 5.

Based on the previously presented analysis, the thermal-comfort plan was classified into four parts: building layout optimisation, square layout optimisation, road layout optimisation, and green space layout optimisation. The sunshade mode of the public space in the village

combines shading from buildings and plants. In addition, the plan includes three new squares (S5, S6, and S7) and two new green spaces (G5 and G6), and the layout of the local roads is improved. The present thermal-comfort map and proposed design plan are shown in Fig 18A and 18B. The details are as follows:

(1) Optimisation of building layout: the building layout can affect the microclimate of the outdoor space and control the wind direction to a certain extent. According to previous studies [93–95], the spatial structure is mainly controlled by the architectural layout. The spacing, orientation, floor height, and shape of a building will also affect the air flow. The aim of this study was to determine a better architectural layout while maintaining the overall architectural layout. The terrain on the south side of the village is high and is currently used as woodland. During the summer, the air flows from the forest to the open area of the village (as such, it is called 'forest wind'). There are two ways to improve the thermal environment. First, the illegal buildings on the south side of the village should be demolished to improve ventilation. Second, the village land should be designed according to the ventilation needs and intensively. Specifically, the buildings on the south side should be reduced in number by transforming the area into a green space. The fewer buildings on the south side can be compensated by increasing the floor height of the buildings on the north side, reducing the local solar radiation intensity.

(2) Optimisation of square layout: the thermal environment of the square is mainly optimised by improving shading, changing the nature of the underlying surface, and increasing spatial openness. More specifically, thermal comfort can be improved ① by adding more green space to the square, creating shaded areas with plants, reducing the direct exposure of villagers to solar radiation in summer, and improving the space utilisation of the square. Furthermore, three squares can be added (S5, S6, S7): S5 is a waterscape square with a shallow water pool in the middle. The waterscape square is surrounded by deciduous trees; the site is equipped with tension film and sunshade. The square S6 is a tree square, and S7 is a basketball hall square. In addition, the S4 square area can be optimised. ② The underlying surface should be replaced to reduce the coverage of the hardened ground of the square, thus reducing the heat stress. In addition to installing permeable tiles in the S1 and S2 squares, the new parking lot in the S7 area is built with grass-planting bricks, which increase the green area and enable the infiltration of rainwater so that sufficient parking space and a cool and comfortable environment are provided for outdoor activities on hot days.

(3) Optimisation of road layout: the comfort of the road space is improved by improving the connectivity of the traffic corridor and increasing the green space next to the road. For example, the road layouts on the left side of R2 and R4 can be changed, and the original layout of the road can be replaced with the north–south layout for better road ventilation. In addition, the road on the east side of the North–South Road of the school can be extended such that the space can be fully exploited. Moreover, street trees can be planted on both sides of the street; the selected deciduous umbrella-shaped trees increase the shading level and create a cool space during the summer.

(4) Optimisation of green space: the green space can be optimised by increasing the green space, reasonably collocating the plant levels, and selecting suitable plant varieties. More specifically, ① the green space should be increased, and broad-leaved trees should be planted. Owing to the good shading effect and leaf transpiration, the temperature under and around broad-leaved trees is relatively low, which improves the thermal environment. Furthermore, the original green space (G3) can be expanded, new residential green space (G5) can be added, and a small garden (G6) can be integrated into the village map. ② the

**Table 5. Thermal comfort level and planning approaches for outdoor public space in villages.**

| Number | Space Type | Specific Geographical Location | Underlying Surface Properties | Plant Configuration | Outdoor Activity Condition | Solutions |
|---|---|---|---|---|---|---|
| 1 | Square | Close to highway | Concrete ground | No | The flow of people is maximal at night. | Replace original hard concrete with permeable paving bricks to ensure rainwater infiltration and self-regulation of thermal environment. |
| 2 | Square | Intersection of West–East Street of Market Road and highway | White tile | No | Low flow of people; residents mainly stay for a short time; no fitness facilities. | Replace paving materials, reduce area with bricks and increase green space. |
| 3 | Road | Middle section of North–South Street of Market Road | Asphalt pavement | The road has plants one sides. | Main roads with high flow of people, primarily because of traffic; there is typically no extended stoppage. | Enlarge green areas next to roads, match plants reasonably and replace tree species with fewer leaves with native plants with higher leaf density index values. |
| 4 | Square | Intersection of North–South Street, Market Road, and highway | Red square bricks | Only a few vines cling to the wooden gallery frame. Solar radiation can still enter the lower space through some green plants during the summer. | Flow of people is concentrated at night; area has fitness facilities. | Increase the square green area, while ensuring ventilation and lighting; in addition, increase the number of shading trees. |
| 5 | Road | Middle section of West–East Street of Market Road | Asphalt pavement | On both sides of the road, the bifurcation point of the street trees is low, and the leaf density is low; the low heights of the plants does not provide sufficient shade. | Main roads with high flow of people and traffic, usually without extended stoppage. | Increase the roadside trees on both sides of the road; replace the trees with low branching points with those with high branching points, and replace trees with low leaf density index values with native plants with higher leaf density index values to enhance the ventilation performance and provide shading on the road during the summer. |
| 6 | Green space (irregular shape) | West side of East–West Street of Market Road | Grassland | Many wild plants block the ventilation space. | Low pedestrian flow; people generally do not stop here. | Increase manual maintenance; enrich plant levels, and improve the openness of space. |
| 7 | Road | North side of Park Road | Asphalt pavement | More shrub species than arbour species on both sides of the road. | Secondary trunk roads with low pedestrian flow; people generally do not stop here. | Plant trees with dense leaves. |
| 8 | Road | Middle section of Maling 5th Group Road | Asphalt pavement | North side of road has low-growing shrubs, and south side has street trees; nevertheless, overall plant spacing is great, and trees are small. | Main access road; the stoppage time is short. | Increase trees on south side of road; reduce tree spacing in some areas to an appropriate distance and replace original tree species with those with high leaf density index values. |
| 9 | Green space (rectangle) | East side of Park Road | Grassland | Middle part of green space constitutes a microtopography with 0.5 m height; plant community with rich natural landscapes and densely planted trees, shrubs, and grass (e.g. Koelreuteria paniculata, Ginkgo biloba, Sophora japonica, Rosa chinensis). | In the morning, special personnel water and maintain the plants in the park. There is a large flow of people in the morning and afternoon, and the main activities are chatting, watching, and walking. | Adjust plant density and some varieties in the middle micro-terrain area, increase ventilation performance of park space and replace road surface material of garden road with permeable pavement. |

(*Continued*)

**Table 5.** (Continued)

| Number | Space Type | Specific Geographical Location | Underlying Surface Properties | Plant Configuration | Outdoor Activity Condition | Solutions |
|---|---|---|---|---|---|---|
| 10 | Green space (polygon) | West side of Park Road | Grassland | Plant species are abundant and mainly densely planted; however, there are more plants with low branch points (e.g. Pinus bungeana, cedar, tower pine, Sabina chinensis, and Prunus cerasifera Ehrh.). | There is a large flow of people in the afternoon. In the morning, special personnel water and maintain the plants in the park. The main activities of the villagers are chatting, watching and walking. | Reduce planting of plants with low branch points and increase ventilation performance of park space. Tree species with dense leaves and high branching points such as Magnolia, purple leaf plum, acacia, sweet-scented Osmanthus, and tree heather can be added; in addition, trees with a better landscape effect such as ginkgo and cherry blossoms can be planted to enrich the landscape level and increase ventilation. |
| 11 | Road | Photovoltaic Road | Asphalt pavement | Distributed on one side of the road. | Secondary trunk roads with low pedestrian flow; people generally do not stop here. | Plant more street trees on both sides of road for more tree shading. |
| 12 | Road | West Street of Primary School | Concrete ground | Walnut and other trees are planted on north side of road, buildings are located on south side, and some illegal buildings block the ventilation effect on the south side of the village. | Secondary trunk road with low flow of people. | Keep original walnut trees on the road, demolish illegal buildings on the south side of the street in order to increase the permeability. |
| 13 | Green space (ribbon) | North side of fishpond | Grassland | Lawn is sparse and unsheltered; area is dominated by ground-cover plants, without trees and shrubs. | Flow of people is maximal in the afternoon. | Plant tree species with high branches (good ventilation), provide space below the tree crowns, and ensure shading while reducing the barrier of the lower layer to the south wind. |
| 14 | Square | Intersection of Photovoltaic Road and North–South Street of School Road | Red square bricks | Plant coverage rate is low, only some low arbours. | High flow of people; there are fitness facilities in the open-air square, and the villagers perform outdoor activities in the morning, noon, and evening; at night, there are more young people. The main activities are morning exercises, parent–child entertainment, and fitness. | Plant deciduous tree species with high leaf density indexes in order to get sunshade in summer and more sunshine in winter. Different changes of four levels of plants, such as Ginkgo biloba, Koelreuteria, Fraxinus chinensis, and Sophora japonica; construct temporary sunshade facilities, such as sunshade umbrellas, to improve the overall landscape and thermal environment of the square. |
| 15 | Road | North–South Street of School Road | Asphalt pavement | Distributed on one side of the road, and west side of the road has low trees and shrubs. | Main roads have a high flow of people, which changes during the day. | Increase the height of some buildings on the east side and replace the trees on the west side of the original road with trees with higher leaf densities. |

different requirements for thermal comfort during the winter and summer should be considered. The proposed plan includes deciduous plants, evergreen plants, and local trees with high branching points. For example, deciduous trees with broad canopies can be planted around the garden road in the green space, and trees, shrubs, and grass land

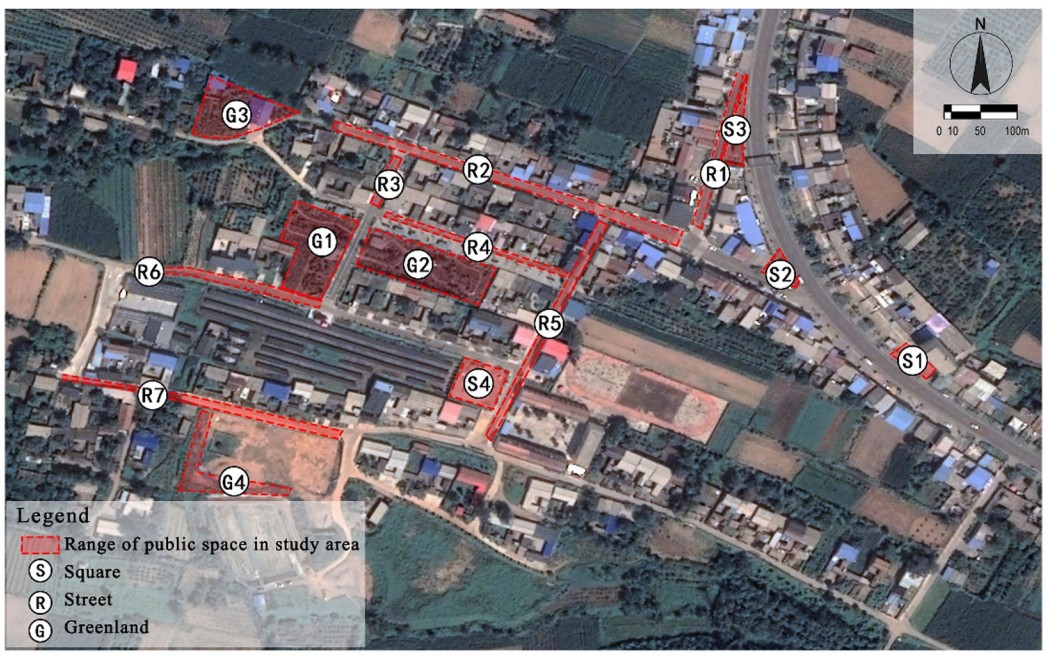

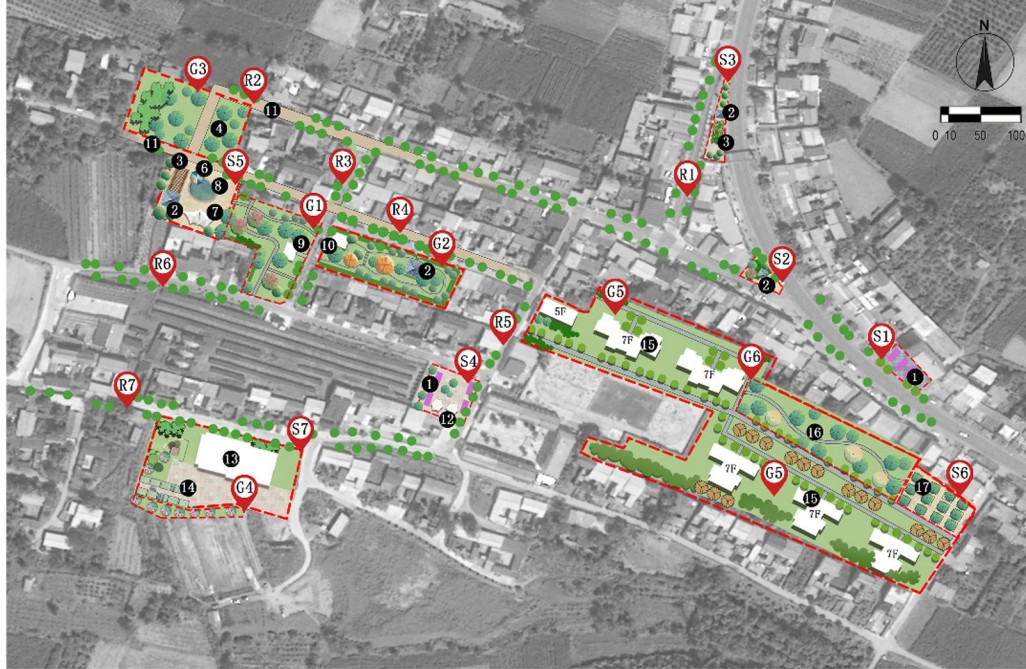

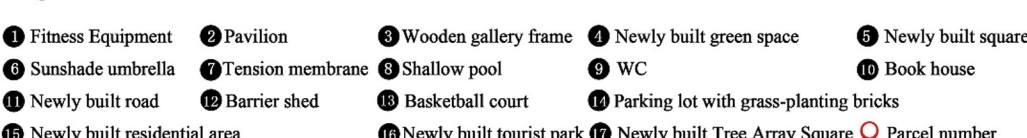

**Fig 18.** (a) Current map of the village's public spaces. (b) Thermal-comfort plan of the village's public spaces. (The satellite images in Fig 18(a) and 18(b) with permission from Beijing Qianfan Shijing Co., Ltd, original copyright [2020] [96]).

should be distributed to properly provide adequately shaded areas in summer and sufficient radiation in winter.

(5) Thermal environment simulation of planned village. To evaluate the proposed plan, PET indicators were used to simulate the planned thermal environment of the village (Fig 19). the neutral temperature of the outdoor public space of the village in summer is 23.77–27.23˚C at 14:00 during the summer. The areas reported to be 'very hot' sensation were outside the boundaries of the village's public spaces. The areas reported to be 'hot' were the main roads, and the areas reported to be 'warm' were the green spaces and squares. Due to the large green area and many tall buildings on the east side, the PET remains low. Therefore, a plan for the thermal environment of the village can improve the overall thermal environment, facilitating the cooling effect on village roads and squares in particular.

## 4. Discussion

(1) Analysis of differences before and after realising the thermal environment plan. Researchers that have studied the thermal environments of urban environments, such as urban squares [60], parks [61], and streets [62], have focused on the measurement and evaluation of indicators. Specific thermal-comfort plans have not been investigated until now. In this study, the influencing factors of the village's thermal environment were studied by evaluating the village's thermal environment and conducting ENVI-met simulations. The resulting, feasible thermal-comfort scheme for the village is presented in Fig 18A. The village's current thermal environment was studied with GIS spatial interpolation (Fig 20) and ENVI-met simulations. The results of the two methods were also compared. The GIS spatial interpolation method reflects the evident variation characteristics of the microclimate parameters at each measurement point. Nevertheless, the method has limitations; some high value points may be neglected in the analysis; for example, Fig 20A presents an area

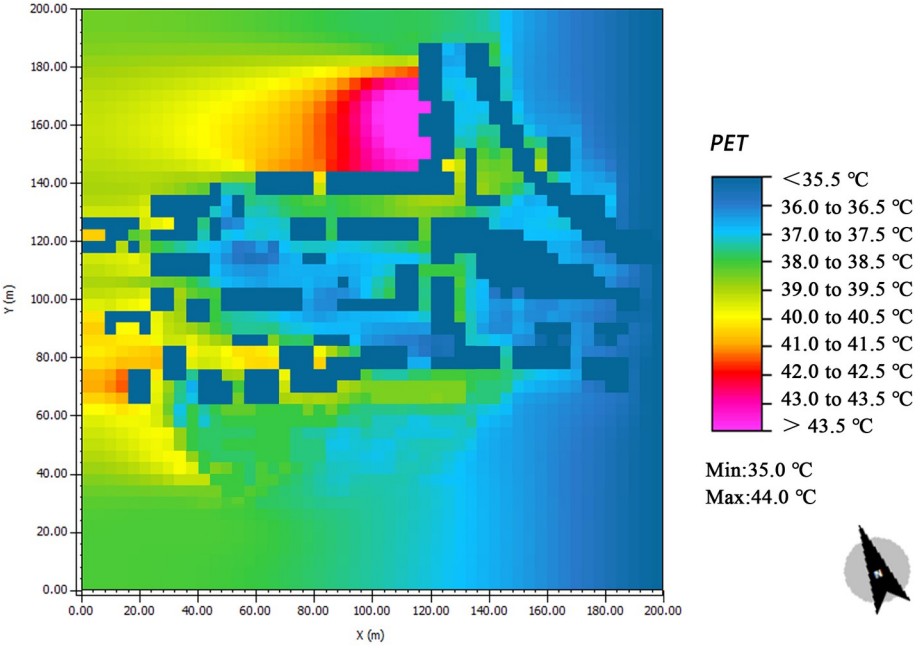

**Fig 19. Spatial distribution of PET in the village.**

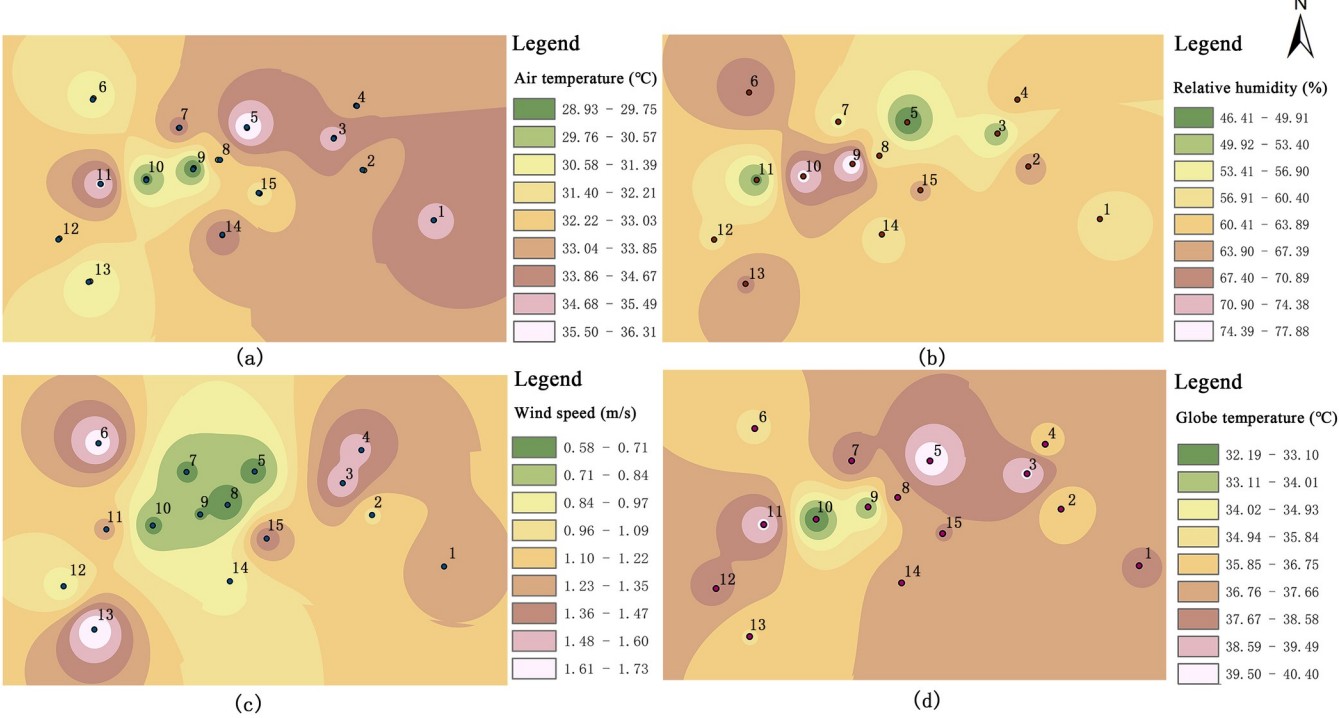

**Fig 20. Spatial distribution map of the thermal environment at each measurement point.**

with a lower temperature between Points 6 and 11; by contrast, the simulation results (Fig 17C) shows an area with a higher temperature between Points 6 and 11. Thus, the software provides more comprehensive results. Moreover, the building height has an evident influence on the wind speed, temperature, solar radiation, and humidity in the village. Nevertheless, the building height must meet the thermal comfort and overall stylistic and cultural requirements of the village. The shortcomings of the spatial analysis method can be overcome by performing additional ENVI-met simulations.

The data from the subjective questionnaire survey and agree well with the objective analysis and reveal some evident problems. For example, the roads and squares have poor thermal comfort conditions, which cause a 'hot' sensation during the summer. Therefore, in the future, more attention should be paid to the thermal comfort conditions of the streets and squares in Maling Village. During the summer, the villagers perform outdoor activities more frequently in cooler areas. A comfortable temperature, humidity, and wind environment can effectively reduce the heat and prolong the residents' outdoor activities. The layout should consider the shapes of the open spaces, plant configurations, nature of the underlying surface, and building heights and layouts as well as other factors.

(2) The surface landscape has a great influence on the thermal comfort of the village based on the theories of urban local climate zone [68–74]. Villages formed different shapes and underlying surface types resulting from different land cover, thus, thermal comfort attributes can be identified based on the surface landscape. Taking Maling as an example, the village was divided into 9 types with different local climate (Fig 21). In future studies, the divided method can be used to distinguish different local climate zones, to evaluate thermal comfort in different zones, and to optimize the village planning according to the microclimate information.

| Instantiation | Category | Description | Instantiation | Category | Description |
|---|---|---|---|---|---|
|  | LCZ 3 Buildings with low-height and high-density. | Mainly rural houses of 1 ~ 3 floors. |  | LCZ 8 Buildings with block shape. | Mainly industrial building with low height. |
|  | LCZ A Dense forests | Mainly forest with high dense of trees. |  | LCZ B Sparse forests | Mainly forest with low dense of trees. |
|  | LCZ C Shrubs | Parks in village |  | LCZ D Low vegetation | Farmland in village |
|  | LCZ E Pavement | Roads and squares |  | LCZ F Naked land | Mainly reserved land |
|  | LCZ G Water | Mainly ponds and rivers |  |  |  |

**Fig 21. Local Climate Zone (LCZ) zoning system in Villages—a case study of Maling.**

## 5. Conclusion

1. There exist significant spatiotemporal regularities of rural thermal environment. In general, the overall thermal comfort conditions were the worst in the village streets, and thus villagers are less willing to stay in these areas for extended periods of times. The green spaces were frequently visited in the morning and afternoon, while the squares were more often visited after supper. The mean air temperature of the East–West Street is higher than that of the North–South Street, the temperature of the green space in the middle of the village is lower than that at the border of the village, and the temperature of the square on the south side of the village is higher than that on the north side. Furthermore, the temperature of the public space increases to its maximum value until 14:00 and remains at that value from 14:00 to 16:00.

2. Different factors have different effects on the thermal comfort of villages. During the summer, the effects of the temperature, solar radiation, wind speed, and relative humidity on the subjective comfort of the village's outdoor space gradually decrease. The factors that influence the thermal comfort level of public spaces are as follows: shape of the public space > plant configuration > building height and layout > nature of underlying surface > street directions > nature of outdoor activity.

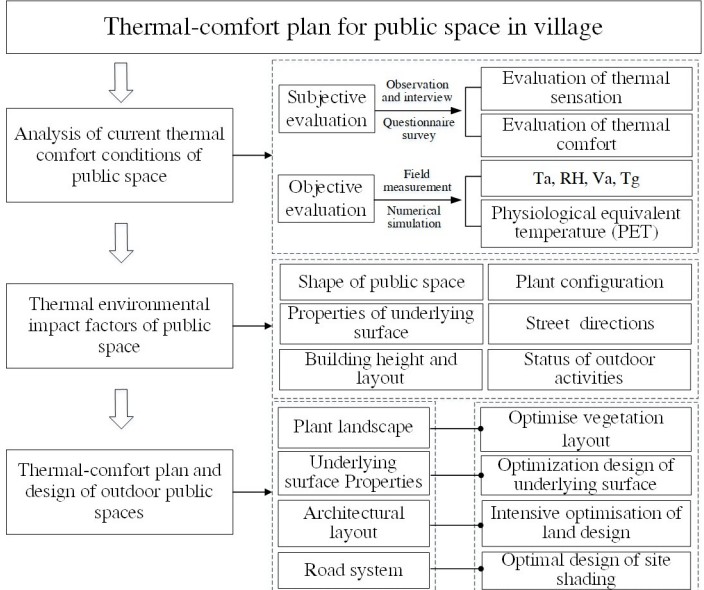

**Fig 22. Thermal-comfort plan for the village's public spaces.**

3. The thermal-comfort planning method (Fig 22) this study proposed is practical. In future studies, the thermal comfort level of public spaces should be considered in the construction layout of the village. A higher thermal comfort level creates a healthier environment, which motivates villagers to perform more outdoor activities.

## Supporting information

**S1 File. Questionnaire used for data collection.**
(DOCX)

**S2 File. Data of subjective evaluation.**
(DOCX)

**S3 File. Data of objective evaluation.**
(XLSX)

## Acknowledgments

First of all. I would like to thank the support of the project funding. I am very grateful to my tutor for helping me to complete the topic selection and conception of the thesis, and gave me a lot of valuable suggestions on the revision of my thesis, so that my paper has been constantly improved. In addition, in the process of completing my thesis, I would like to express my heartfelt thanks to my family and friends who care and help me.

## Author Contributions

**Conceptualization:** Qindong Fan.

**Data curation:** Fengtian Du.

**Formal analysis:** Fengtian Du.

**Funding acquisition:** Qindong Fan, Hu Li.

**Investigation:** Fengtian Du.

**Methodology:** Fengtian Du.

**Project administration:** Qindong Fan, Chenming Zhang.

**Resources:** Qindong Fan.

**Software:** Fengtian Du.

**Supervision:** Qindong Fan, Hu Li, Chenming Zhang.

**Validation:** Fengtian Du.

**Visualization:** Fengtian Du.

**Writing – original draft:** Fengtian Du.

**Writing – review & editing:** Qindong Fan, Fengtian Du, Chenming Zhang.

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
