## [Decision Letter · Decision Letter 0]

7 Apr 2021

PONE-D-21-04373

Thermal-Comfort Evaluation of and Plan for Public Space of Maling Village, Henan, China

PLOS ONE

Dear Dr. DU,

Thank you for submitting your manuscript to PLOS ONE. After careful consideration, we feel that it has merit but does not fully meet PLOS ONE’s publication criteria as it currently stands. Therefore, we invite you to submit a revised version of the manuscript that addresses the points raised during the review process.

We look forward to receiving your revised manuscript.

Kind regards,

Jun Yang

Academic Editor

PLOS ONE

Journal Requirements:

2. Please provide additional details regarding participant consent. In the ethics statement in the Methods and online submission information, please ensure that you have specified (1) whether consent was informed and (2) what type you obtained (for instance, written or verbal, and if verbal, how it was documented and witnessed). If the need for consent was waived by the ethics committee, please include this information.”

3. Please include additional information regarding the survey or questionnaire used in the study and ensure that you have provided sufficient details that others could replicate the analyses. For instance, if you developed a questionnaire as part of this study and it is not under a copyright more restrictive than CC-BY, please include a copy, in both the original language and English, as Supporting Information."

4. Thank you for including your ethics statement:  "The field studies were conducted at an village public open space in Changdai Town, Mengjin County, Luoyang City, Henan Province. Thus, we could conduct experiments there without specific permits. The experiments conducted in this study did not involve endangered or protected species.".

5. Please provide additional details regarding participant consent. In the ethics statement in the Methods and online submission information, please ensure that you have specified (a) whether consent was informed and (b) what type you obtained (for instance, written or verbal, and if verbal, how it was documented and witnessed). If your study included minors, state whether you obtained consent from parents or guardians. If the need for consent was waived by the ethics committee, please include this information.

6. We note that Figures 1, 2 and 15 in your submission contain map/satellite images which may be copyrighted. All PLOS content is published under the Creative Commons Attribution License (CC BY 4.0), which means that the manuscript, images, and Supporting Information files will be freely available online, and any third party is permitted to access, download, copy, distribute, and use these materials in any way, even commercially, with proper attribution. For these reasons, we cannot publish previously copyrighted maps or satellite images created using proprietary data, such as Google software (Google Maps, Street View, and Earth). For more information, see our copyright guidelines: http://journals.plos.org/plosone/s/licenses-and-copyright.

a) You may seek permission from the original copyright holder of Figure(s) [#] to publish the content specifically under the CC BY 4.0 license. 

7. PLOS requires an ORCID iD for the corresponding author in Editorial Manager on papers submitted after December 6th, 2016. Please ensure that you have an ORCID iD and that it is validated in Editorial Manager. To do this, go to ‘Update my Information’ (in the upper left-hand corner of the main menu), and click on the Fetch/Validate link next to the ORCID field. This will take you to the ORCID site and allow you to create a new iD or authenticate a pre-existing iD in Editorial Manager. Please see the following video for instructions on linking an ORCID iD to your Editorial Manager account: https://www.youtube.com/watch?v=_xcclfuvtxQ

9. We note that your supporting information includes an image of a participant in the study. 

Additional Editor Comments (if provided):

Reviewer 1

Overall, this paper is interesting and the measurement and analysis is sound. It has the potential to be accepted after authors revisions (I have done some in the attachment)

1. The English should be improved.

2. The introduction should be improved.

3. The methods should be restructured.

The specific comments are given in the attachment.

Reviewer 2

After a very careful reading of the work entitled "Thermal-Comfort Evaluation of and Plan for Public Space of Maling Village, Henan, China", I have found a very well-done work, well presented and organized, clear in concepts and methodology. The topic and context attract attention for many readers from various disciplines. The study is worth to be published in PLOS ONE after conducting the revisions.

1. Local climate zone is a very meaningful research. I think the author can apply this idea to this study, so as to improve the applicability of the results.

Suggest reading, for instance,

https://doi.org/10.1109/JSTARS.2018.2808469

https://doi.org/10.1016/j.scs.2019.101487

https://doi.org/10.1016/j.jclepro.2020.123767

https://doi.org/10.1016/j.uclim.2020.100700

https://doi.org/10.1016/j.scs.2021.102818

2. The language of this manuscript needs to be revised by native English speaking experts.

3.The format of the paper should be consistent

Reviewers' comments:

Reviewer's Responses to Questions

**Comments to the Author**

1. Is the manuscript technically sound, and do the data support the conclusions?

Reviewer #1: Yes

Reviewer #2: Yes

2. Has the statistical analysis been performed appropriately and rigorously? 

Reviewer #1: Yes

Reviewer #2: Yes

3. Have the authors made all data underlying the findings in their manuscript fully available?

Reviewer #1: Yes

Reviewer #2: Yes

4. Is the manuscript presented in an intelligible fashion and written in standard English?

Reviewer #1: Yes

Reviewer #2: No

5. Review Comments to the Author

Reviewer #1: Overall, this paper is interesting and the measurement and analysis is sound. It has the potential to be accepted after authors revisions (I have done some in the attachment)

1. The English should be improved.

2. The introduction should be improved.

3. The methods should be restructured.

The specific comments are given in the attachment.

Reviewer #2: After a very careful reading of the work entitled "Thermal-Comfort Evaluation of and Plan for Public Space of Maling Village, Henan, China", I have found a very well-done work, well presented and organized, clear in concepts and methodology. The topic and context attract attention for many readers from various disciplines. The study is worth to be published in PLOS ONE after conducting the revisions.

1. Local climate zone is a very meaningful research. I think the author can apply this idea to this study, so as to improve the applicability of the results.

Suggest reading, for instance,

https://doi.org/10.1109/JSTARS.2018.2808469

https://doi.org/10.1016/j.scs.2019.101487

https://doi.org/10.1016/j.jclepro.2020.123767

https://doi.org/10.1016/j.uclim.2020.100700

https://doi.org/10.1016/j.scs.2021.102818

2. The language of this manuscript needs to be revised by native English speaking experts.

3.The format of the paper should be consistent.

6. PLOS authors have the option to publish the peer review history of their article (what does this mean?). If published, this will include your full peer review and any attached files.

Reviewer #1: No

Reviewer #2: No

---

## [Author Response · Author response to Decision Letter 0]

22 Jul 2021

1.Question 1

Response:

Thank you for your suggestion. We have carefully revised the format of the manuscript in accordance with the requirements of the journal strictly, including those for file naming.

2.Question 2 and 5: Ethics statement

Response:

The field studies were conducted in Maling village, Changdai Town, Mengjin County, Luoyang City, Henan Province, China. The public open space of the village was taken as the study area. This study was supported by the local government and the masses. The questionnaire survey and field measurement did not involve endangered or protected species. For a detailed explanation, please see the Supporting Information.

3.Question 3: Questionnaire investigation

Response:

Outdoor Thermal comfort questionnaire

Date: / /____ Time: __________ 

1 Gender 

Male Female 

Height: _________ Weight: _________

2 What is your age group range?

 18-29 years 

 30-40 years 

 41-50 years 

 51-60 years 

 61-64 years

65years and older

3 What are you wearing now? (multiple choice)

 Pale colors Neutral Dark colors

Upper: Sleeveless vest Short-sleeve T-shirt 

 Long- sleeve T-shirt Long- sleeve blouse Sweater 

Bottom: Short shorts Straight trousers (thin) long skirt

Short skirt Other_______

4 Location: Space with fitness facilities Unsheltered square 

 Spaces with shading from trees Unshaded lawns Green park spaces Pavilions Areas with shading from buildings Roads

5 Please describe your overall comfort level?

（Note: Please vote according to your actual situation at this time）

6 How do you feel at this moment?

7 Your current activity?

Standing (chatting, playing mobile phone, enjoy scenery, etc.)

Seating (chatting, playing mobile phone, reading, etc.)

Strolling

Low-intensity exercising (brisk walking, looking after children, walk the dog, etc.)

Medium-intensity exercising (Jogging, etc.)

 High- intensity exercising (ball games, square dancing, etc.)

8 How long have you been here?

＜15 min 15-30 min 30-60 min ＞60 min

9 What time do you like to come here during the day?

6:00-8:00 8:00-10:00 10:00-12:00 12:00-14:00 14:00-16:00 16:00-18:00 18:00-20:00 20:00-21:00

10 Which meteorological parameters do you think have the greatest impact on thermal comfort? (select two of them)

Air temperature Relative humidity Wind speed Solar radiation 

11 You will be more comfortable if the environment (single choice for each)

Air temperature: Higher Unchanged Lower

Relative humidity: Damper Unchanged Drier

Wind speed: Stronger Unchanged Weaker 

Solar radiation： Stronger Unchanged Weaker

室外热舒适问卷调查

日期: / /____ 时间: __________ 

1 性别 

男 女 

身高: _________ 体重: _________

2 您处于哪个年龄阶段?

 18-29 岁 

 30-40 岁 

 41-50 岁 

 51-60 岁 

 61-64 岁

65岁及以上

3 您现在穿什么? (多选)

 白色 中性色 黑色

上部: 无袖背心 短袖T恤 长袖T恤 

 长袖衬衫 毛衣

下部: 超短裤 直裤（薄） 长裙

短裙 其他_______

4位置: 有健身设施的地方 无遮挡的广场 遮阳树下 

 无遮阳的草坪 公园绿地 凉亭 

 建筑物的阴影区 道路

5 请描述一下您的整体舒适度？

（注：请根据本次实际情况投票）

6 此时此刻你感觉如何?

7 您目前的活动?

站立 (聊天, 玩手机, 欣赏风景等)

坐着 (闲聊, 玩手机, 阅读等)

散步

低强度运动 快步走, 照看孩子, 遛狗等)

中强度运动 (慢跑等)

 高强度运动 (球类运动,广场舞等)

8 您在这里多久了?

＜15 分钟 15-30 分钟 30-60分钟 ＞60分钟

9 在一天中，您喜欢什么时候来这里？

6:00-8:00 8:00-10:00 10:00-12:00 12:00-14:00

14:00-16:00 16:00-18:00 18:00-20:00 20:00-21:00

10 您认为哪些气象参数对热舒适的影响最大？ (选择其中的两个)

空气温度 相对湿度 风速 太阳辐射 

11 如果环境是……你会感觉更舒适（每类选择一个）

空气温度: 更高 不变 更低

相对湿度: 更潮湿 不变 更干

风速: 更强 不变 更弱 

太阳辐射： 更强 不变 更弱

4.Question 4: Please amend your current ethics statement to include the full name of the ethics committee/institutional review board(s) that approved your specific study

Response:

The full name of the institution that approved our research is "people's Government of Changtai Town, Mengjin County".

5.Question 6:

The satellite images in Figures 1, 2 and 17 (a) in this study are all from Beijing Qianfan Shijing Co., Ltd, and the contract is signed as follows:

6.Question 7:

The ORCID iD of all authors are as follows:

FAN QINDONG, http://orcid.org/0000-0003-2370-6252

DU FENGTIAN, https://orcid.org/0000-0003-4879-2409

LI HU, http://orcid.org/0000-0003-3416-8250

ZHANG CHENMING, https://orcid.org/0000-0001-7579-3285

7.Question 8:

We have attached the captions of the supporting information files at the end of the manuscript, and update any in-text citations to match accordingly.

For a detailed explanation, please see the Supporting Information.

8.Question 9:

We've removed the images about the participants in the Supporting Information. For a detailed explanation, please see the Supporting Information.

---

## [Decision Letter · Decision Letter 1]

9 Aug 2021

Thermal-Comfort Evaluation of and Plan for Public Space of Maling Village, Henan, China

PONE-D-21-04373R1

Dear Dr. DU,

We’re pleased to inform you that your manuscript has been judged scientifically suitable for publication and will be formally accepted for publication once it meets all outstanding technical requirements.

Kind regards,

Jun Yang

Academic Editor

PLOS ONE

Additional Editor Comments (optional):

Accept

Reviewers' comments:

Reviewer's Responses to Questions

**Comments to the Author**

1. If the authors have adequately addressed your comments raised in a previous round of review and you feel that this manuscript is now acceptable for publication, you may indicate that here to bypass the “Comments to the Author” section, enter your conflict of interest statement in the “Confidential to Editor” section, and submit your "Accept" recommendation.

Reviewer #1: All comments have been addressed

Reviewer #2: All comments have been addressed

2. Is the manuscript technically sound, and do the data support the conclusions?

Reviewer #1: Yes

Reviewer #2: Yes

3. Has the statistical analysis been performed appropriately and rigorously? 

Reviewer #1: Yes

Reviewer #2: Yes

4. Have the authors made all data underlying the findings in their manuscript fully available?

Reviewer #1: Yes

Reviewer #2: Yes

5. Is the manuscript presented in an intelligible fashion and written in standard English?

Reviewer #1: Yes

Reviewer #2: Yes

6. Review Comments to the Author

Reviewer #1: I believe authors have already well addressed all my concerns and questions. I suggest the acceptance of this paper.

Reviewer #2: The quality of this manuscript has been significantly improved. Some problems in the first edition have been revised according to the comments of the reviewers. The study is worth publishing.

7. PLOS authors have the option to publish the peer review history of their article (what does this mean?). If published, this will include your full peer review and any attached files.

Reviewer #1: No

Reviewer #2: No

---

## [Editor Report · Acceptance letter]

3 Sep 2021

PONE-D-21-04373R1 

Thermal-Comfort Evaluation of and Plan for Public Space of Maling Village, Henan, China 

Dear Dr. Du:

I'm pleased to inform you that your manuscript has been deemed suitable for publication in PLOS ONE. Congratulations! Your manuscript is now with our production department. 

Kind regards, 

on behalf of

Dr. Jun Yang 

Academic Editor

PLOS ONE